# Cohesin-interacting protein WAPL-1 regulates meiotic chromosome structure and cohesion by antagonizing specific cohesin complexes

Oliver Crawley[†], Consuelo Barroso, Sarah Testori, Nuria Ferrandiz, Nicola Silva[‡], Maikel Castellano-Pozo, Angel Luis Jaso-Tamame, Enrique Martinez-Perez*

Meiosis group, MRC Clinical Sciences Centre, Faculty of Medicine, Imperial College London, London, United Kingdom

**Abstract** Wapl induces cohesin dissociation from DNA throughout the mitotic cell cycle, modulating sister chromatid cohesion and higher-order chromatin structure. Cohesin complexes containing meiosis-specific kleisin subunits govern most aspects of meiotic chromosome function, but whether Wapl regulates these complexes remains unknown. We show that during *C. elegans* oogenesis WAPL-1 antagonizes binding of cohesin containing COH-3/4 kleisins, but not REC-8, demonstrating that sensitivity to WAPL-1 is dictated by kleisin identity. By restricting the amount of chromosome-associated COH-3/4 cohesin, WAPL-1 controls chromosome structure throughout meiotic prophase. In the absence of REC-8, WAPL-1 inhibits COH-3/4-mediated cohesion, which requires crossover-fated events formed during meiotic recombination. Thus, WAPL-1 promotes functional specialization of meiotic cohesin: WAPL-1-sensitive COH-3/4 complexes modulate higher-order chromosome structure, while WAPL-1-refractory REC-8 complexes provide stable cohesion. Surprisingly, a WAPL-1-independent mechanism removes cohesin before metaphase I. Our studies provide insight into how meiosis-specific cohesin complexes are regulated to ensure formation of euploid gametes.

*For correspondence: enrique. martinez-perez@imperial.ac.uk

**Present address:** [†]Department of Neuroscience, The Scripps Research Institute, Jupiter, United States; [‡]Max F. Perutz Laboratories, University of Vienna, Vienna, Austria

**Competing interests:** The authors declare that no competing interests exist.

## Introduction

Structural maintenance of chromosome (SMC) proteins take part in complexes that associate with DNA to promote key events of the cell cycle, such as chromosome condensation and segregation, DNA repair, and gene expression (*Jeppsson et al., 2014*). The cohesin complex, which mediates sister chromatid cohesion (SCC) between S-phase and chromosome segregation at anaphase (*Michaelis et al., 1997*), consists of two SMC proteins (Smc1 and Smc3) plus a kleisin subunit (Scc1/Rad21), forming a tripartite structure that topologically embraces DNA molecules (*Haering et al., 2002*; *Haering et al., 2008*). A fourth cohesin subunit (Scc3) binds to the kleisin and is also required for the functionality of the complex, while other proteins associate temporarily with cohesin to regulate its binding to DNA (*Haarhuis et al., 2014*). Cohesin is loaded to chromosomes by the Scc2/4 complex (*Ciosk et al., 2000*), and SCC is established during DNA replication in a process that involves acetylation of Smc3 (*Ben-Shahar et al., 2008*; *Unal et al., 2008*; *Zhang et al., 2008*). SCC is ultimately dissolved at anaphase onset, when cleavage of the kleisin subunit by the protease separase triggers the segregation of sister chromatids to opposite poles of the spindle (*Uhlmann et al., 1999*). Proper establishment and release of SCC is also essential for chromosome segregation during meiosis, the specialized cell division program that produces haploid gametes from diploid germ cells (*Petronczki et al., 2003*).

**eLife digest** Most of the genetic material of plant and animal cells is stored in structures called chromosomes. Nearly all the cells in the body contain two copies of each chromosome, one inherited from the mother and the other from the father, but sex cells – such as egg and sperm – contain just one copy of each. If eggs or sperm contain the wrong number of copies of a chromosome, genetic disorders such as Down syndrome can occur.

New sex cells form in a process called meiosis, which begins with a cell that contains two copies of each chromosome duplicating each of these copies. The duplicated copies are known as sister chromatids, and are held together by a ring-like protein complex called cohesin. In addition to tethering sister chromatids, cohesin affects the 'higher-order' organization of chromosome structure and promotes the recruitment of other proteins that are essential for different aspects of chromosome behavior during meiosis. Therefore, regulating cohesin binding during meiosis is key to ensuring that sex cells contain the correct number of chromosomes.

Cohesin is ultimately removed from chromosomes in two steps during the consecutive cell divisions at the end of meiosis, resulting in the formation of sex cells containing a single copy of each chromosome. However, whether cohesin is actively removed from chromosomes during early meiosis, when chromosomes undergo dramatic structural changes, is not known.

Using a combination of microscopy and genetic techniques to study the developing egg cells of the worm *Caenorhabditis elegans*, Crawley et al. investigated how a protein called WAPL-1 affects cohesin binding to chromosomes during early meiosis. This revealed that WAPL-1's effects depend on the identity of a particular subunit of the cohesin complex. If this subunit is a protein called COH-3 or COH-4, WAPL-1 reduces the ability of cohesin to bind to chromosomes during the early stages of meiosis. However, WAPL-1 does not affect cohesin complexes that instead feature a protein called REC-8 as this subunit.

By preventing excessive binding of COH-3 and COH-4 cohesin, WAPL-1 regulates chromosome structure and sister chromatid cohesion during early meiosis. Crawley et al. further observed that during the stage preceding the first meiotic division, cohesin is removed from chromosomes by a mechanism that does not involve WAPL-1.

The next challenge is to work out why cohesin containing the REC-8 protein is protected from being released by WAPL-1. Whether defects in this protection can trigger the premature separation of sister chromatids is also an important question to answer.

In addition to the separase-dependent removal of cohesin at anaphase onset, a pathway dependent on the Wapl protein removes cohesin from chromosomes at earlier stages of the cell cycle in somatic cells (*Gandhi et al., 2006*; *Kueng et al., 2006*). Wapl is thought to destabilize the interaction between the kleisin and Smc3 subunits, allowing the release of cohesin from DNA without catalytically cleaving any subunit (*Chan et al., 2012*; *Eichinger et al., 2013*; *Huis in 't Veld et al., 2014*). Cohesin complexes in which the Smc3 subunit is acetylated during DNA replication become resistant to Wapl and remain stably bound to DNA, thereby providing persistent SCC (*Ben-Shahar et al., 2008*; *Nishiyama et al., 2010*; *Lopez-Serra et al., 2013*). However, acetylated cohesin is also removed from chromosome arms during prophase and early prometaphase in mammalian cells, when phosphorylation of Scc3 and Sororin, a protein that antagonizes Wapl, renders these complexes sensitive to Wapl (*Hauf et al., 2005*; *Nishiyama et al., 2010*; *Nishiyama et al., 2013*). This mode of cohesin removal is known as the prophase pathway and its failure in cells lacking Wapl causes increased arm cohesion in metaphase chromosomes and defects in chromosome segregation (*Waizenegger et al., 2000*; *Haarhuis et al., 2013*; *Tedeschi et al., 2013*). Removal of Wapl before S-phase causes a large increase in chromatin-associated cohesin and dramatic changes in chromosome organization (*Tedeschi et al., 2013*), demonstrating that Wapl is a key regulator of SCC and chromatin organization throughout the mitotic cell cycle.

Cohesin is an essential component of meiotic chromosomes, not only by mediating SCC, but also by promoting the acquisition of structural features required for meiotic chromosome function (*McNicoll et al., 2013*). Meiotic chromosomes are organized as linear arrays of chromatin loops,

which are attached at their base to cohesin-containing proteinaceous axial elements (*Kleckner, 2006*). Proper assembly of axial elements during early prophase is required for subsequent pairing and recombination between homologous chromosomes. Crossovers formed during recombination, together with SCC, form attachments between homologous chromosomes (chiasmata) that are responsible for the correct orientation of chromosomes on the first meiotic spindle and ultimately for their correct partitioning during the meiotic divisions (*Petronczki et al., 2003*). Crucially, these events require the formation of axial elements containing meiosis-specific cohesin complexes in which the mitotic kleisin Scc1 is substituted by Rec8 (*Klein et al., 1999*; *Watanabe and Nurse, 1999*). Moreover, additional meiosis-specific kleisins beyond Rec8 have been identified in mouse (Rad21L) (*Herrán et al., 2011*; *Ishiguro et al., 2011*; *Lee and Hirano, 2011*) and in *C. elegans*, where the highly homologous and functionally redundant COH-3 and COH-4 kleisins associate with SMC-1 and SMC-3 to form cohesin complexes that associate with meiotic chromosomes independently of REC-8 cohesin (*Severson et al., 2009*; *Severson and Meyer, 2014*). Although Rad21L and COH-3/4 are not essential for SCC, these kleisins are required for pairing and recombination between homologous chromosomes (*Ishiguro et al., 2014*; *Severson and Meyer, 2014*). Interestingly, large amounts of Rad21L and COH-3/4 are removed from chromosomes before metaphase I (*Herrán et al., 2011*; *Ishiguro et al., 2011*; *Lee and Hirano, 2011*; *Severson and Meyer, 2014*), and a prophase pathway has recently been proposed to operate during late meiotic prophase in plants (*De et al., 2014*). However, whether Wapl induces cohesin removal at any stage of meiotic prophase in animals, and whether Wapl may regulate some of the functions of different meiosis-specific cohesin complexes is not known.

Using the *C. elegans* germ line, which contains a complete time course of meiotic prophase, we demonstrate that WAPL-1 antagonizes cohesin binding from the onset of meiosis, and show that cohesin complexes containing the COH-3/4 kleisins are specifically targeted by WAPL-1. By antagonizing the binding of COH-3/4 complexes to axial elements, WAPL-1 acts as a regulator of meiotic chromosome structure and SCC. Moreover, we also show that SCC is modulated by WAPL-1 and recombination during the chromosome remodeling process that starts at the end of pachytene, and report that a WAPL-1-independent mechanism removes cohesin during the oocyte maturation process preceding metaphase I.

## Results

### WAPL-1 is required for fertility

In order to investigate the role of WAPL-1 during meiotic prophase, we used a deletion allele, *wapl-1(tm1814)*, that removes the first two exons of the *C. elegans Wapl* homolog (*Figure 1A*). Western blot analysis on whole-worm protein extracts showed that the WAPL-1 protein is absent in *wapl-1 (tm1814)* mutants, confirming that the *tm1814* deletion is a null allele of *wapl-1* (*Figure 1B*). Homozygous *wapl-1(tm1814)* mutants (referred from now on as *wapl-1* mutants) are viable, but display a reduction in brood size and high levels of embryonic lethality (*Figure 1C*). In addition to these reproductive defects, *wapl-1* mutants also displayed somatic defects, as demonstrated by the high incidence of larval arrest amongst the hatched *wapl-1* embryos (*Figure 1C*) and by the presence of an egg laying defect in adult worms. In order to prevent the accumulation of somatic defects, all the analysis presented here was performed in homozygous *wapl-1* worms derived from heterozygous mothers.

The overall organization of *wapl-1* mutant germ lines appears largely normal, with clearly defined mitotic and meiotic compartments in which the different stages of meiotic prophase can be easily identified (*Figure 1D*). In fact, observation of diakinesis oocytes (the last stage of meiotic prophase) showed that both wild type and *wapl-1* mutant oocytes displayed 6 DAPI-stained bodies, demonstrating that WAPL-1 is not required for chiasma formation (*Figure 1E*). Nonetheless, the reduced fertility of *wapl-1* mutants suggested that WAPL-1 may play important roles in the germ line. Thus, we investigated the staining pattern of WAPL-1 during meiosis by creating transgenic worms homozygous for the *tm1814* deletion and for a single-copy insertion of a transgene that expresses a GFP:: WAPL-1 fusion protein using the 5' and 3' UTRs from the *wapl-1* locus. Expression of this transgene largely rescued the fertility defects of *wapl-1(tm1814)* mutants (*Figure 1C*), and western blot analysis confirmed the presence of a band of the expected molecular weight for the GFP::WAPL-1 fusion

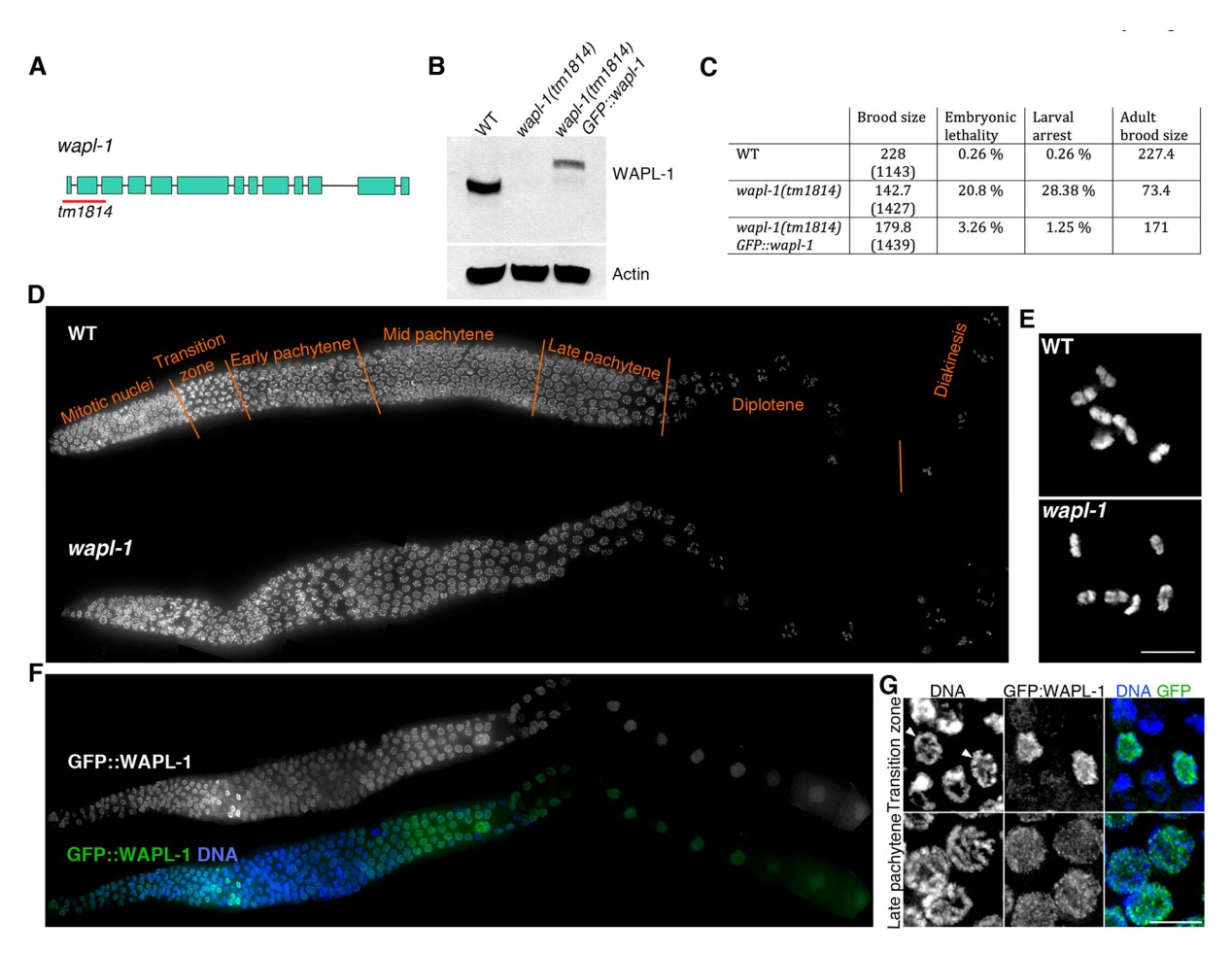

**Figure 1.** WAPL-1 localizes to germ line nuclei and promotes viability. (**A**) Structure of the *wapl-1* gene, red bar indicates the region deleted in the *tm1814* allele. (**B**) Western blot demonstrates that *wapl-1(tm1814)* is a null allele and that a protein of the expected size is present in worms carrying a *GFP::wapl-1* transgene. (**C**) *wapl-1* mutants display reduced fertility and larval lethality, numbers in parenthesis indicate total number of embryos analysed per genotype . (**D**) Projections of whole-mounted germ lines stained with DAPI, the different stages of meiotic prophase are noted above the WT germ line, with transition zone containing nuclei in leptotene and zygotene. Note that overall germ line organization in *wapl-1* mutants is similar to WT. (**E**) Projections of diakinesis oocytes stained with DAPI, six bivalents are present in both WT and *wapl-1* mutants. (**F**) Whole-mounted germ line from a transgenic worm homozygous for the *wapl-1(tm1814)* deletion and for a *GFP::wapl-1* single copy transgene stained with DAPI and anti-GFP antibodies. Note that the intensity of GFP::WAPL-1 decreases in transition zone and peaks again during late pachytene. (**G**) Insets from germ line shown in F showing GFP::WAPL-1 staining in transition zone and pachytene nuclei, note that GFP::WAPL-1 intensity is very high in transition zone nuclei that do not display chromosome clustering (arrowheads). *Figure 1—figure supplement 1* shows quantification of GFP::WAPL-1 intensities along the germ line. Scale bars in E and G = 5 µm.

The following figure supplement is available for figure 1:

**Figure supplement 1.** Quantification of GFP::WAPL-1 intensity.

protein, although the overall intensity of this band was reduced compared to the endogenous WAPL-1 protein (*Figure 1B*). Staining of germ lines from these transgenic worms with anti-GFP antibodies demonstrated that the GFP::WAPL-1 protein is present, with a diffuse staining pattern, in both mitotic and meiotic nuclei (*Figure 1F*). Interestingly, the intensity of the GFP::WAPL-1 signal decreased drastically as transition zone nuclei acquired the chromosome clustering characteristic of early meiotic prophase stages (leptotene and zygotene), peaking again at late pachytene and then remaining at similar high levels in diplotene and diakinesis oocytes (*Figure 1F–G* and *Figure 1—figure supplement 1*).

## WAPL-1 promotes timely repair of meiotic DSBs and correct polar body extrusion during the meiotic divisions

Despite normal formation of chiasmata, the high incidence of embryonic lethality among the progeny of *wapl-1* mutants (*Figure 1C*) suggested the existence of meiotic defects. Moreover, the presence of developmental defects among the progeny of *wapl-1* mutants could be a consequence of defects in DNA repair during meiosis, so we monitored the progression of meiotic recombination by visualizing the appearance and disappearance of the RAD-51 recombinase, which labels early meiotic recombination intermediates (*Colaiácovo et al., 2003*), and of COSA-1 and ZHP-3 foci, two proteins that are required for crossover formation and that localize specifically to crossover-fated recombination events in late pachytene nuclei (*Bhalla et al., 2008*; *Yokoo et al., 2012*). We observed 6 COSA-1 and 6 ZHP-3 foci in both *wapl-1* mutants and wild-type controls (*Figure 2A* and *Figure 2—figure supplement 1*), consistent with the normal presence of chiasmata in *wapl-1* diakinesis oocytes. Despite this, RAD-51-positive recombination intermediates accumulate in mid pachytene nuclei of *wapl-1* mutants (*Figure 2B*). Furthermore, while RAD-51 foci were no longer detected in 98% of late pachytene nuclei of wild-type controls, 52% percent of nuclei in the same region of *wapl-1* mutant germ lines displayed RAD-51 foci. Thus, although crossover precursors are successfully formed, the repair of a subset of DSBs is delayed in *wapl-1* mutants.

Next, we investigated if the meiotic divisions proceeded normally in the absence of WAPL-1, as defects in this process could induce aneuploidy even if chiasma formation is not affected. During oogenesis, each meiotic division results in the formation of a polar body that contains a full complement of homologs (meiosis I) or sister chromatids (meiosis II). Polar bodies are extruded away from the egg pronucleus, localizing on the cortex and not contributing to the genetic content of the developing embryo (*Figure 3A*). All *wapl-1* mutant embryos analyzed formed two polar bodies, however, only 22% of post meiotic embryos (up to the two-cell embryo) displayed both polar bodies at the cortex, with most embryos (61%) displaying one polar body located away from the cortex and the remaining (11%) displaying both polar bodies away from the cortex (*Figure 3A–B*). In addition, we noticed that chromatin morphology in the polar bodies was altered in *wapl-1* mutants, often displaying separated DNA masses within the polar body (*Figure 3—figure supplement 1*). To gain better understanding of the effect of WAPL-1 on polar body extrusion, we performed live imaging of the meiotic and early mitotic divisions in wild type and *wapl-1* mutant embryos expressing histone H2B::mcherry (*Figure 3D; Videos 1–3*). These experiments confirmed that most *wapl-1* mutant embryos display defects in the extrusion of the second polar body (*Figure 3C–D*; *Videos 2–3*). We also observed that in 2 out of 11 *wapl-1* mutant embryos, the second polar body, which failed to migrate to the cortex, underwent chromatin decondesation and condensation cycles, mimicking the changes observed in the mitotic nuclei of the embryo (*Figure 3D* and *Videos 2–3*). Moreover, in 1 out of 11 filmed embryos the second polar body eventually fused with one of the mitotic nuclei generated after the first mitotic division (*Figure 3D* and *Video 3*), demonstrating that the failure in polar body extrusion of *wapl-1* mutants can lead to aneuploidy in the embryo. We also used fluorescence in situ hybridization (FISH) to investigate if chromosome non-disjunction occurs during the meiotic divisions of *wapl-1* mutants. Labeling of the 5S rDNA locus on chromosome V in early embryos demonstrated that the oocyte pronucleus contained one single signal for the 5S rDNA in all embryos analyzed from wild-type (17) and *wapl-1* mutants (17) (*Figure 3E*), suggesting that chromosome V segregates properly during the meiotic divisions in the absence of WAPL-1. This analysis demonstrates that WAPL-1 is required to ensure proper polar body extrusion during the meiotic divisions, and suggests that defects in this process may contribute to the embryonic lethality observed in *wapl-1* mutants.

## WAPL-1 regulates axial element morphogenesis at meiosis onset and axis compaction during pachytene

During the initial observation of *wapl-1* mutant germ lines we noticed that pachytene chromosomes appeared more widely spaced within the nucleus and somewhat thicker than in wild-type controls, suggesting that WAPL-1 may regulate the shape of meiotic chromosomes, as it does in mitotic cells (*Tedeschi et al., 2013*). Meiotic chromosomes are organized around axial elements containing cohesin and meiosis-specific HORMA-domain proteins (*Kleckner, 2006*). Thus, we used antibodies against HTP-3, a HORMA-domain protein that is an essential component of axial elements in *C.*

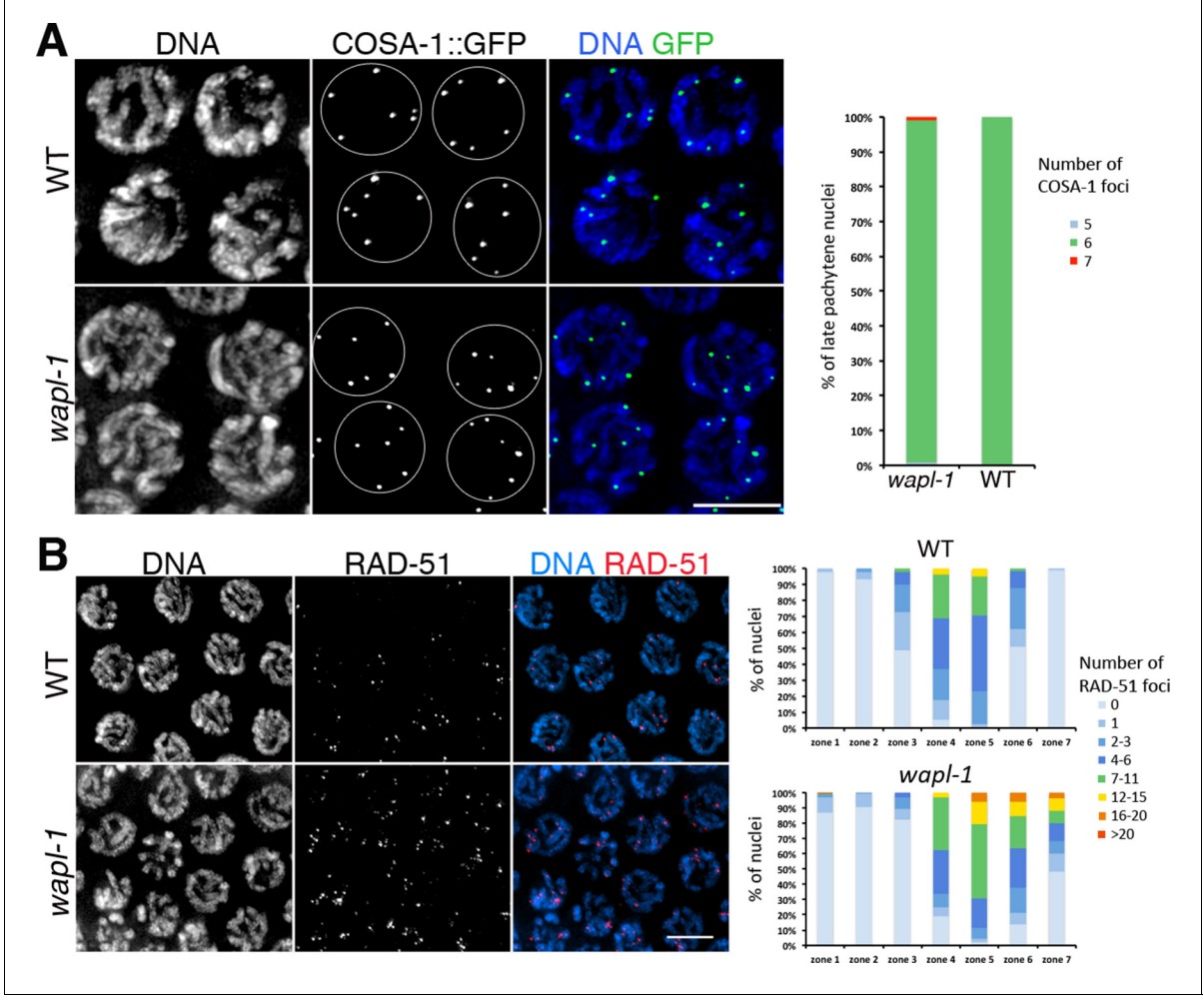

**Figure 2.** WAPL-1 affects DNA repair during meiotic prophase. (**A**) Projections of late pachytene nuclei from worms expressing COSA-1::GFP, note that both *wapl-1* mutants and WT controls display 6 COSA-1 foci per nucleus. Graph showing quantification of COSA-1 foci (126 nuclei from *wapl-1* mutants and 100 nuclei from WT). (**B**) Projections of pachytene nuclei stained with anti-RAD-51 antibodies and DAPI, note increased RAD-51 foci in *wapl-1* mutant panel. Quantification of RAD-51 foci in germ lines of WT and *wapl-1* mutants. Each germ line was divided into 7 equal-sized regions, with regions 4 to 7 representing early to late pachytene. The X axis indicates the seven regions along the germ line, while the Y axis indicates the percentage of nuclei with a given number of RAD-51 foci (as indicated in the color key). *wapl-1* mutants accumulate RAD-51 in mid and late pachytene nuclei. Number of nuclei analyzed (WT, *wapl-1* mutant): Zone 1 (133, 138), zone 2 (246, 195), zone 3 (137, 135), zone 4 (154, 101), zone 5 (122, 90), zone 6 (114, 69), zone 7 (93, 61).

The following figure supplement is available for figure 2:

**Figure supplement 1.** *wapl-1* mutants form normal numbers of ZHP-3 foci.

*elegans* (*Goodyer et al., 2008*; *Severson et al., 2009*), to investigate chromosome organization in *wapl-1* mutants. Projections made from late pachytene nuclei of wild-type germ lines demonstrated ample overlap between different HTP-3-labeled axial elements, making it difficult to follow individual HTP-3 tracks along their whole length (*Figure 4A* and *Figure 4—figure supplement 1*). In contrast, the overlap of HTP-3 tracks was reduced in *wapl-1* mutants, with some nuclei displaying six distinctive HTP-3 tracks (one per homolog pair) that could be clearly traced along their full length (*Figure 4A* and *Figure 4—figure supplement 1*). Measuring of total HTP-3 track length per nucleus demonstrated a 28% decrease in axial element length in late pachytene nuclei of *wapl-1* mutants (*Figure 4B*).

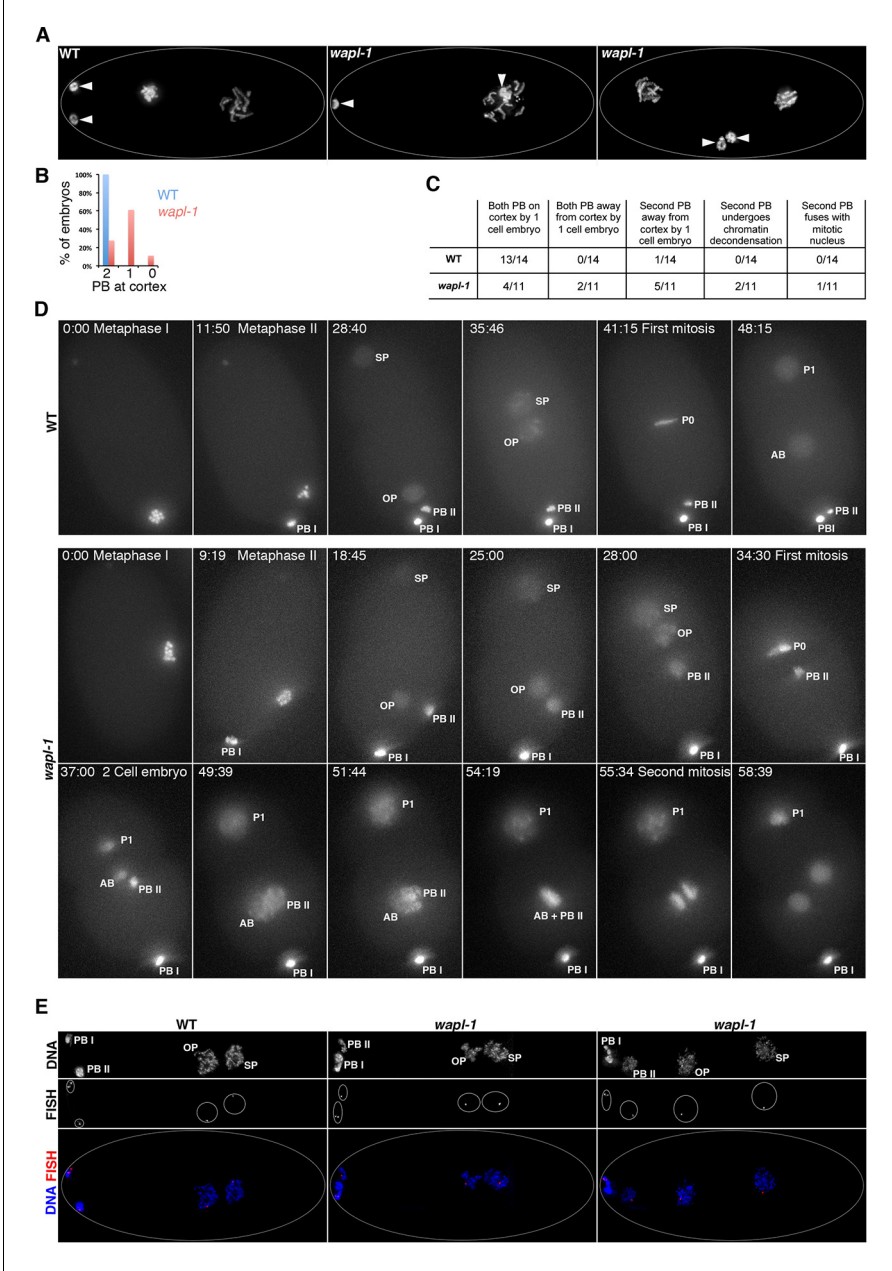

**Figure 3.** WAPL-1 is required for polar body extrusion. (**A**) Projections of fixed embryos at the 1- or 2-cell stage stained with DAPI, arrowheads point to the position of polar bodies generated during the meiotic divisions. Polar bodies are found near the cortex in WT control, but one (middle panel) or both (right-hand side panel) polar bodies localize away from the cortex in *wapl-1* mutant embryos. (**B**) Quantification of the percentage of embryos with zero, one, or two polar bodies localized at the cortex (36 embryos scored in *wapl-1* and 34 in WT). (**C**) Quantification of polar body behavior in videos from live WT and *wapl-1* mutant embryos expressing a histone H2B::mcherry fusion protein. Note that 7 out of 11 *wapl-1* mutant embryos displayed defects in polar body extrusion. Examples of videos used for the quantification are shown in **Video 1** (WT), **Videos 2–3** (*wapl-1*). (**D**) Selected frames from the WT embryo shown in **Video 1** and from the *wapl-1* mutant embryo shown in **Videos 2–3**. Time is indicated on top-left corner, starting from metaphase I. Abbreviations: PB I (first polar body), PB II (second polar body), OP (oocyte pronucleus), SP (sperm pronucleus), P0 (first mitotic metaphase following fusion of OP and SP), P1 and AB (cells resulting from the first mitotic division). Note that in the WT embryo PB II remains highly condensed and locates close to PB I on the cortex. In the *wapl-1* mutant embryo, PB II becomes decondensed and fails to move to the cortex, first remaining close to the OP and then close to the AB cell produced after the first mitotic division. Chromosomes from AB and PB II appear to mix together before the

*Figure 3 continued on next page*

*Figure 3 continued*

second mitotic division of the embryo. (**E**) Projections of fixed embryos following the completion of the second meiotic division and labeled with a FISH probe against the 5S rDNA locus on chromosome V and DAPI. Note that in both WT and *wapl-1* mutant embryos the oocyte pronucleus (OP) and the second polar body (PB II) contain a single FISH signal, even when PB II is not localized on the cortex and chromatin appears decondensed (*wapl-1* example on right-hand side).

The following figure supplement is available for figure 3:

**Figure supplement 1.** Example of abnormal chromatin condensation in polar bodies of *wapl-1* mutant embryos.

---

The organization of meiotic chromosomes in pachytene nuclei is not only determined by axial elements, but also by the synaptonemal complex (SC), a proteinaceous structure that glues together homologous axial elements to promote stable pairing and inter-homolog recombination (*MacQueen et al., 2002*). In order to understand better the effect of WAPL-1 on the organization of axial elements, we investigated chromosome structure in pachytene nuclei of *wapl-1 syp-1* double mutants, since lack of SYP-1 prevents SC assembly without affecting the formation of axial elements (*MacQueen et al., 2002*). Late pachytene nuclei of *syp-1* mutant germ lines displayed many long and thin HTP-3 tracks, corresponding to individualized and elongated axial elements (*Figure 4C*). In contrast, late pachytene nuclei of *wapl-1 syp-1* double mutants contained fewer HTP-3 tracks, which also appeared bulkier (*Figure 4C*). This suggested that axial elements were much shorter, and/or the presence of some kind of SC-independent association between axial elements, in *wapl-1 syp-1* double mutants. Visualization of the 5S rDNA locus on chromosome V by FISH showed that this region was unpaired in most pachytene nuclei of both *syp-1* and *wapl-1 syp-1* double mutants, demonstrating that increased

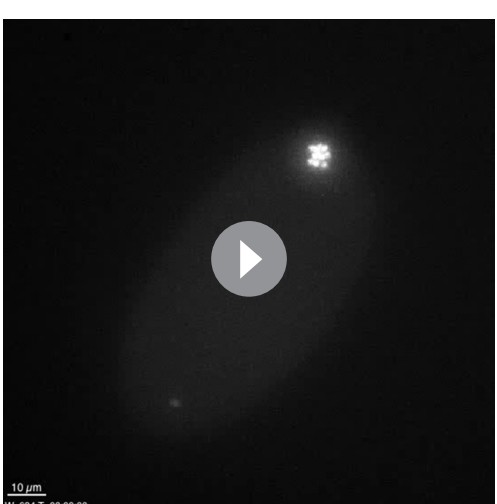

**Video 1.** Live imaging of a WT embryo expressing histone H2B::mcherry. Filming covers the interval between the first meiotic metaphase and the end of the first mitotic division. Each meiotic division results in the production of a polar body and both polar bodies remain highly condensed and located at one end of the embryo, away from the oocyte and sperm pronuclei. *Figure 3D* contains individual images from this video in which specific meiotic and mitotic events are labeled (images in *Figure 3D* are rotated with respect to the video).

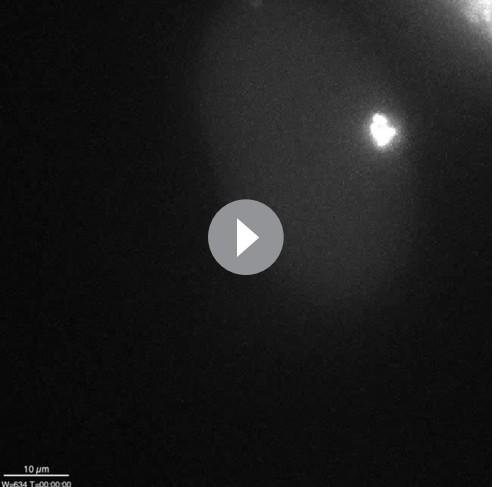

**Video 2.** Live imaging of a *wapl-1* mutant embryo expressing histone H2B::mcherry. Filming covers the interval between the first meiotic division and the end of the first mitotic division. Note that the second polar body does not migrate to the cortex, instead it follows the movement of the oocyte pronucleus towards the middle of the embryo and the chromosomes appear decondensed. As chromosomes in the sperm and oocyte pronuclei condense in preparation for the first mitotic division, condensation of chromatin also occurs in the second polar body. *Figure 3D* contains individual images from this video in which specific meiotic and mitotic events are labeled, and *Video 3* shows continued filming from the same embryo.

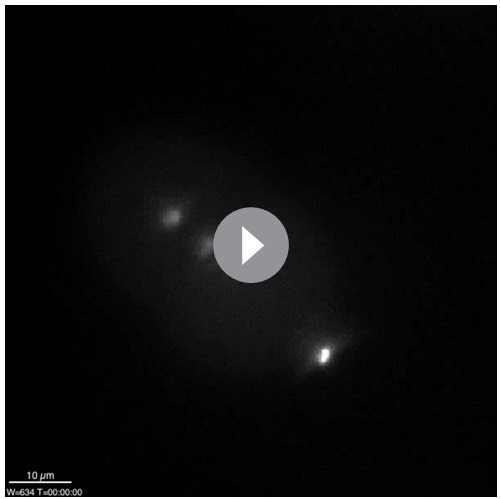

**Video 3.** Continuation of live imaging of the *wapl-1* mutant embryo shown in *Video 2*. Filming covers the interval between the end of the first mitotic division and up to the four cell embryo. Following the completion of the first mitotic division, chromatin in the second polar body (PB II) undergoes decondensation. PB II localizes to the vicinity of the AB mitotic nucleus, and as chromosomes in the AB nucleus condense so do chromosomes in PB II. At this point condensed chromosomes from the AB nucleus and PB II appear to mix up before dividing into two daughter nuclei. *Figure 3D* contains individual images from this video in which specific mitotic events are labeled.

association of homologous axial elements is not responsible for the overall reduction in total axial element length observed in *wapl-1 syp-1* mutants (*Figure 4D* and *Figure 4—figure supplement 2*). Moreover, visualization of the X chromosome pairing center region using HIM-8 antibodies (*Phillips et al., 2005*) demonstrated that at late pachytene, when this region becomes separated in *syp-1* mutants (*MacQueen et al., 2002*) (*Figure 4—figure supplement 2*), each HIM-8 signal is associated with a short track of axial element in *syp-1 wapl-1* double mutants (*Figure 4E* and *Figure 4—figure supplement 3*). Thus, removal of WAPL-1 induces dramatic changes in the overall organization of axial elements in pachytene nuclei.

Since cohesin loading during early meiosis is an essential step in the assembly of axial elements (*Severson et al., 2009*; *Lightfoot et al., 2011*; *Llano et al., 2012*), and Wapl antagonizes stable cohesin binding before S-phase in mitotic cells (*Tedeschi et al., 2013*), we investigated whether WAPL-1 regulates the morphogenesis of axial elements at meiosis onset. In order to visualize all cohesin complexes in the germ line we used CRISPR to add a C-terminal GFP tag on the endogenous *smc-1* gene. Homozygous *smc-1:: GFP* worms are viable and healthy and, as expected, the SMC-1::GFP protein is present in all germline and somatic nuclei. To clearly define the onset of meiotic prophase (leptotene stage),

we stained germ lines of SMC-1::GFP wild type and *wapl-1* mutant worms with PLK-2 antibodies, as PLK-2 forms aggregates on the nuclear envelope of leptotene nuclei to promote homolog pairing (*Labella et al., 2011*). In wild-type germ lines, the presence of elongated SMC-1::GFP structures (axial elements) coincided with the appearance of PLK-2 aggregates (8 out 8 germ lines), while nuclei preceding the formation of PLK-2 aggregates displayed diffuse SMC-1::GFP staining (*Figure 4F* and *Figure 4—figure supplement 4*). In contrast, the appearance of elongated SMC-1::GFP structures preceded the formation of PLK-2 aggregates in transition zone nuclei of *wapl-1* mutant germ lines (7 out of 8 germ lines, average of 5 nuclei per germ line) (*Figure 4F*). Interestingly, transition zone nuclei in which chromosome clustering has not yet occurred display the highest intensity of WAPL-1 staining in the whole germ line (*Figure 1F–G* and *Figure 1—figure supplement 1*), suggesting that high levels of WAPL-1 activity may be present in these nuclei. These observations suggest that WAPL-1 regulates the timing of axial element assembly by antagonizing cohesin association with chromosomes during early meiosis.

## WAPL-1 regulates chromosome remodeling at the end of pachytene

During late pachytene meiotic chromosomes initiate a major remodeling process that includes the ordered disassembly of the SC, chromosome condensation and changes in axial element composition (*Chan et al., 2004*; *Nabeshima et al., 2005*; *de Carvalho et al., 2008*; *Martinez-Perez et al., 2008*). Whether the early steps of this process involve major changes in the association of cohesin with chromosomes is not known, but imaging of SMC-1::GFP in wild-type germ lines demonstrated that a diffuse pool of nuclear SMC-1::GFP starts accumulating in late pachytene, becoming very prominent in diplotene and diakinesis oocytes (*Figure 4G* and *Figure 4—figure supplement 5*). This staining pattern suggests that there is strong expression of *smc-1* during late prophase, but that most of this SMC-1 remains unbound to chromosomes, and/or that cohesin is actively removed from axial elements during these stages. In contrast, a diffuse pool of SMC-1::GFP is not observed in

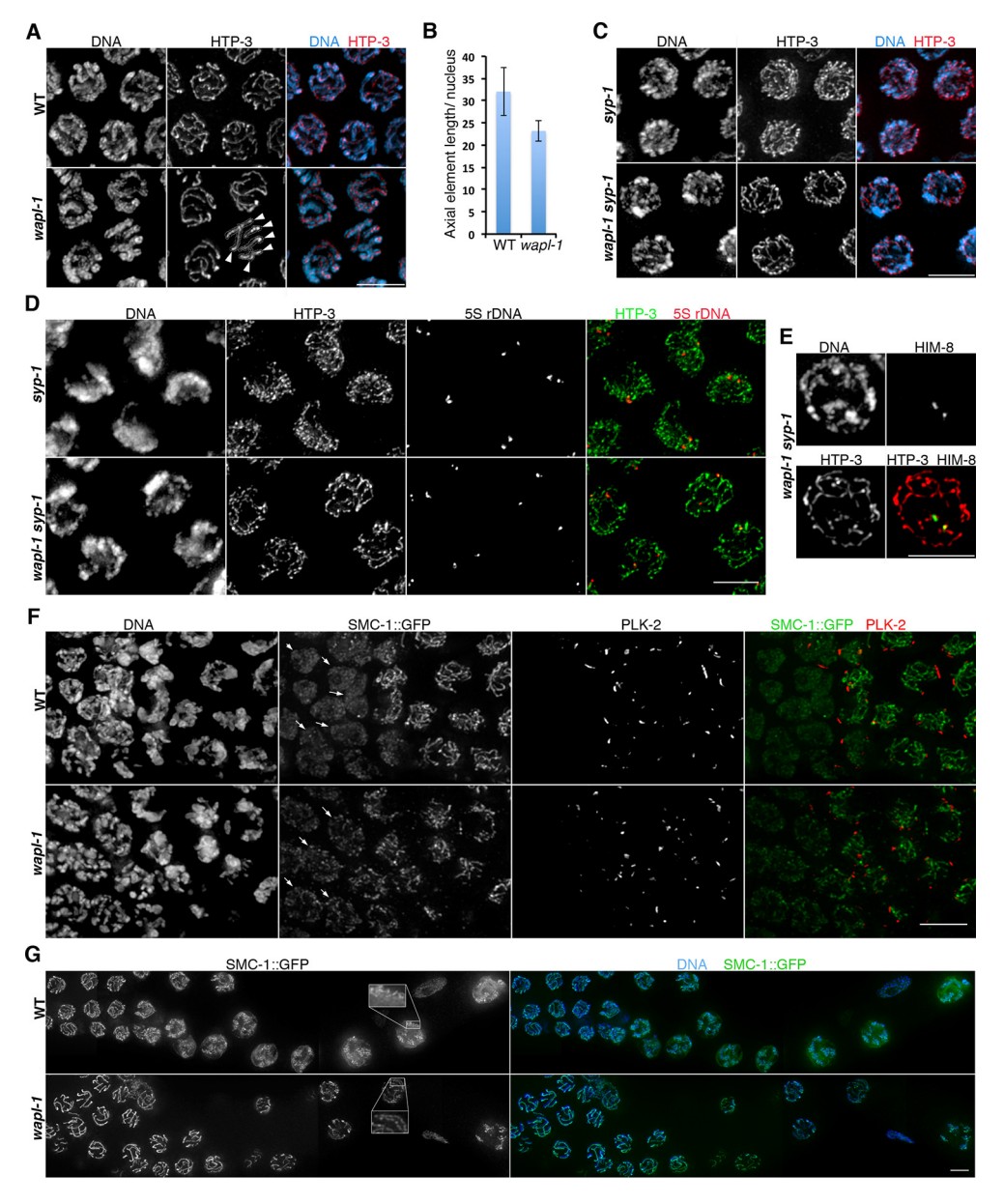

**Figure 4.** WAPL-1 regulates chromosome organization during meiotic prophase. (**A**) Projections of late pachytene nuclei stained with anti-HTP-3 antibodies (axial elements) and DAPI. Axial elements are shorter in *wapl-1* mutants, arrowheads point to 6 HTP-3 tracks that can be individually traced along their whole length (each track represents a pair of aligned homologous chromosomes). (**B**) Quantification of total HTP-3 length per late pachytene nucleus in WT controls and *wapl-1* mutants. Error bars represent standard deviation, differences are significant (*p*<0.0001, *t*-test). (**C**) Projections of late pachytene nuclei stained with HTP-3 antibodies (axial elements) and DAPI. The number of HTP-3 tracks appears larger in *syp-1* mutants than in *wapl-1 syp-1* double mutants, where some short HTP-3 tracks are seen. (**D**) Projections of mid pachytene nuclei stained with anti-HTP-3 antibodies (axial elements), DAPI, and labeled with a probe against the 5S rDNA locus on chromosome V. Most nuclei in *syp-1* and *wapl-1 syp-1* mutants display 2 5S foci, showing that homologs are not associated (quantification shown in *Figure 4— figure supplement 2*). (**E**) Projection of a late pachytene nucleus stained with DAPI, anti-HTP-3 antibodies and anti-HIM-8 antibodies (binding to a single end of the X chromosome). Both HIM-8 signals are located at the end of a short HTP-3 track, each one representing a highly compacted X chromosome. Shortening of X chromosome axial elements in late pachytene nuclei was seen in 3 out of 3 *syp-1 wapl-1* germ lines. (**F**) Projections of transition zone nuclei from worms carrying a GFP tag on the endogenous *smc-1* gene (generated by CRISPR) stained with DAPI, anti-GFP antibodies, and anti-PLK-2 antibodies. Appearance of PLK-2 aggregates on the nuclear envelope

*Figure 4 continued on next page*

*Figure 4 continued*

marks the onset of meiotic prophase (leptotene stage). In the WT germ line, SMC-1::GFP tracks are only observed in nuclei with PLK-2 aggregates, while pre-leptotene nuclei display diffuse SMC-1::GFP staining (arrows). SMC-1:: GFP tracks are present in pre-leptotene nuclei of *wapl-1* mutants (arrows). (**G**) Projections of late pachytene and diplotene nuclei from worms carrying a GFP tag on the endogenous *smc-1* gene (generated by CRISPR) stained with DAPI and anti-GFP antibodies. A large accumulation of nuclear soluble SMC-1::GFP is present in wild-type nuclei, but not in *wapl-1* mutants (see quantification on *Figure 4—figure supplement 5*). Note that axial elements become elongated, twisted and with a more diffuse appearance in wild-type nuclei compared with *wapl-1* mutant nuclei (insets show magnification of the indicated nuclear region). Scale bar = 5 μm in all panels.

The following figure supplements are available for figure 4:

**Figure supplement 1.** Examples of HTP-3 tracking in projections of late pachytene nuclei stained with anti-HTP-3 antibodies.

**Figure supplement 2.** Quantification of homolog pairing in germ lines of *syp-1* and *wapl-1 syp-1* double mutants.

**Figure supplement 3.** Shortening of X chromosome axial elements in *wapl-1 syp-1* double mutants.

**Figure supplement 4.** WAPL-1 affects nuclear organization during early meiosis.

**Figure supplement 5.** WAPL-1 induces accumulation of nuclear soluble SMC-1::GFP in diplotene nuclei.

**Figure supplement 6.** SC disassembly is delayed in *wapl-1* mutant germ lines.

late prophase nuclei of *wapl-1* mutants (*Figure 4G* and *Figure 4—figure supplement 5*). Moreover, while axial elements labeled by SMC-1::GFP remained as easily traceable linear structures in diplotene nuclei of *wapl-1* mutants, they appeared as longer and more diffuse structures in wild-type oocytes (*Figure 4G*). This suggests that WAPL-1 antagonizes cohesin binding during late pachytene and diplotene stages. Interestingly, the intensity of nuclear GFP::WAPL-1 undergoes an increase during late pachytene and diplotene compared with earlier prophase (*Figure 1F* and *Figure 1—figure supplement 1*), suggesting that WAPL-1 activity may be increased during these stages.

In addition to the increased amount of cohesin associated with diplotene chromosomes, *wapl-1* mutants display other defects in the chromosome remodeling process. First, diplotene nuclei in *wapl-1* mutants do not show the coiling of axial elements that is characteristic of this stage in wild-type germ lines (*Nabeshima et al., 2005*) and instead axial elements visualized by SMC-1::GFP or anti-HTP-1 antibodies display a more linear appearance (*Figure 4G* and *Figure 4—figure supplement 6*). Second, *wapl-1* mutants display altered SC disassembly, as evidenced by the persistence of long tracks of SC component SYP-1 in diplotene oocytes (*Figure 4—figure supplement 6*). These observations suggest that cohesin removal by WAPL-1 is an intrinsic feature of chromosome remodeling during late pachytene and diplotene.

## WAPL-1 regulates the levels of chromosome-bound COH-3/4 cohesin complexes, but not REC-8 complexes

The evidence presented above strongly suggests that *wapl-1* mutants undergo meiotic prophase with increased levels of axis-associated cohesin. During *C. elegans* meiosis cohesin complexes carrying different kleisin subunits, either REC-8 or the highly homologous and functionally redundant COH-3 and COH-4, associate with axial elements to ensure efficient SCC and crossover formation (*Pasierbek et al., 2001*; *Severson et al., 2009*; *Severson and Meyer, 2014*). Thus, we used anti-REC-8 and anti-COH-3/4 antibodies (*Figure 5—figure supplement 1*) to determine the staining pattern of the different meiotic cohesin complexes in the presence and absence of WAPL-1. Upon initial observation, the intensity of REC-8 staining appeared very similar in *wapl-1* mutant germ lines and wild-type controls (*Figure 5A*). However, the COH-3/4 signal was clearly increased in *wapl-1* mutants (*Figure 5A*). A quantitative analysis of the mean nuclear intensity of REC-8 and COH-3/4 staining between leptotene and late pachytene confirmed that *wapl-1* mutant germ lines display increased

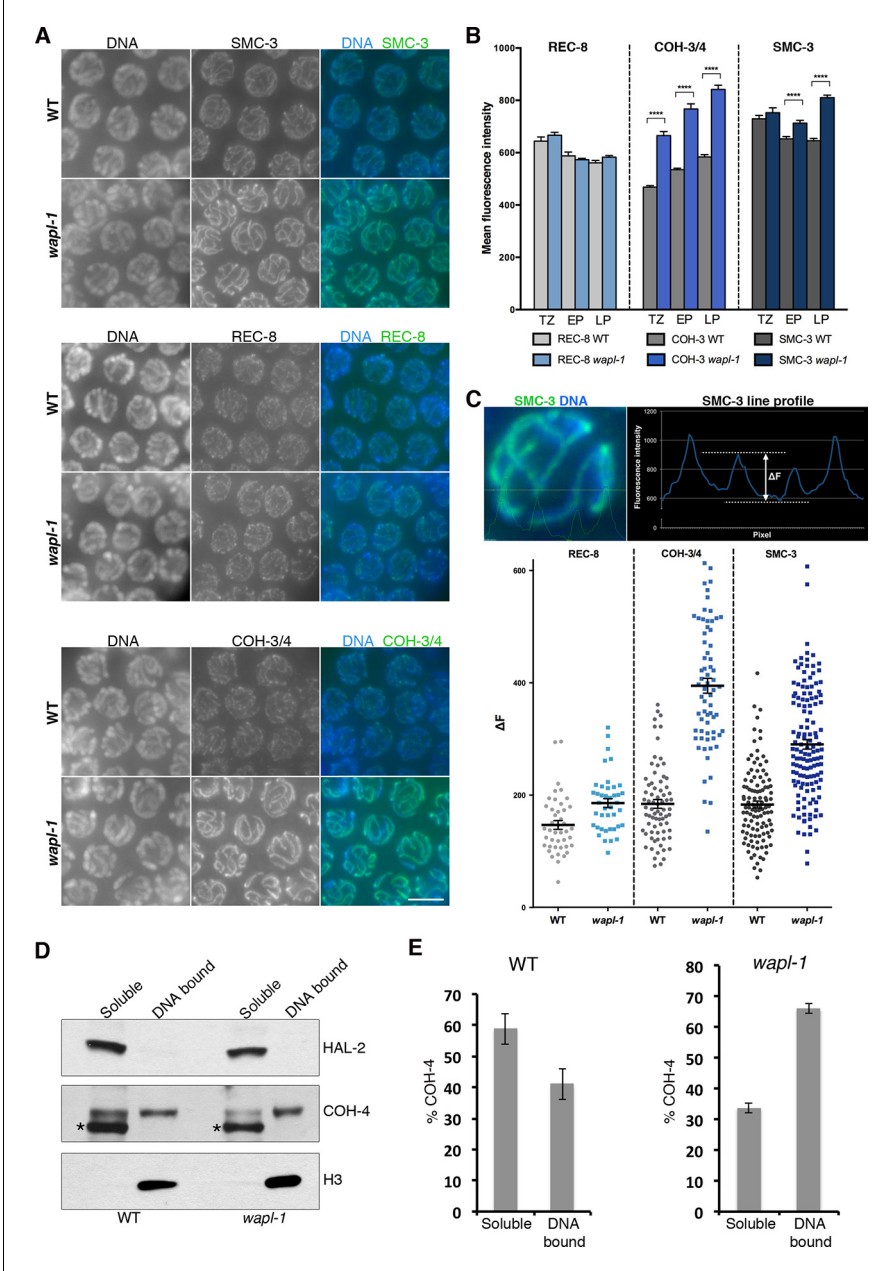

**Figure 5.** WAPL-1 antagonizes binding of COH-3/4 cohesin to axial elements. (**A**) Projections of pachytene nuclei stained with the indicated antibodies and DAPI. In all cases, WT and *wapl-1* examples were acquired with the same exposure settings and images are non-deconvolved projections adjusted with same settings to allow visual comparisons in staining intensity. The intensities of SMC-3 and COH-3/4 are increased in *wapl-1* mutants, while REC-8 staining appears similar in WT and *wapl-1* nuclei. Scale bar = 5 µm. (**B**) Quantification of mean fluorescence intensity per nucleus of REC-8, COH-3/4 and SMC-3 in transition zone (TZ), early pachytene (EP) and late pachytene (LP) nuclei. Between 15 and 20 nuclei per germ line, from a minimum of 5 germ lines were analyzed per genotype and stage. Differences indicated with asterisks are significant (*p*<0.0001, *t*-test), error bars= SEM. (**C**) Line profile quantification to compare the intensity of cohesin at axial elements versus inter-chromosome domains. Top left-hand side panel: example of a SMC-3 and DAPI-stained pachytene nucleus showing the intensities of DAPI (blue) and SMC-3 (green) along the depicted line. The line profile of SMC-3 intensity is shown in the top right-hand panel, ΔF indicates the increment in staining between the peak (axial element) and the valley (inter chromosome domain as determined by lack of DAPI staining). Graph: Plotting of individual ΔF values from late pachytene nuclei of *wapl-1* mutants and WT stained with REC-8, COH-3/4 or SMC-3 antibodies. Between 12 and 32 nuclei from different germ lines were analyzed per genotype. Mean value and SEM are indicated. Proportional

*Figure 5 continued on next page*

*Figure 5 continued*

increase between the mean ΔF value in *wapl-1* and WT: 27% for REC-8 (*p*= 0.0008, *t*-test), 114% for COH-3/4 (*p*<0.0001, *t*-test), 59% for SMC-3 (*p*<0.0001, *t*-test). (D) Western blots of triton soluble and insoluble (DNA bound) protein fractions from WT and *wapl-1* mutant worms probed with anti-COH-4 (see *Figure 5—figure supplement 5* for additional controls), anti-HAL-2 (marker for soluble fraction), and anti-H3 (marker for DNA-bound fraction) antibodies. Asterisk indicates a non-specific band recognized by anti-COH-4 antibodies (see *Figure 5—figure supplement 5*). Note that COH-4 signal in WT extracts is higher in the soluble than in the DNA fraction, while in *wapl-1* mutant extracts COH-4 intensity is higher in DNA-bound than in the soluble fraction. (E) Quantification of relative intensity of anti-COH-4 signal in the soluble and DNA-bound fractions. Three westerns were included in the analysis.

The following figure supplements are available for figure 5:

**Figure supplement 1.** Controls showing specificity of anti-COH-3/4 antibodies.

**Figure supplement 2.** A different anti-COH-3 antibody also shows increased staining intensity in pachytene nuclei of *wapl-1* mutant germ lines.

**Figure supplement 3.** A REC-8::GFP transgene shows similar staining intensity in pachytene nuclei of WT and *wapl-1* mutant germ lines.

**Figure supplement 4.** Controls demonstrating the functionality of the REC-8::GFP transgene.

**Figure supplement 5.** Control western blots for anti-COH-4 antibodies.

---

levels of COH-3/4 throughout meiotic prophase, with the highest differences detected in late pachytene nuclei (*Figure 5B*). The same results were obtained using a COH-3-specific antibody, instead of the COH-3/4 used above, and anti-GFP antibodies to determine the staining of REC-8::GFP expressed from a single copy, and fully functional, transgene (*Figure 5—figure supplements 2–4*). In order to confirm that removal of WAPL-1 causes an increase in the levels of axis-associated COH-3/4, we also performed a line profile analysis of REC-8, COH-3/4 and SMC-3 intensity across late pachytene nuclei. This analysis allows a direct comparison of the signal intensity associated with axial elements versus the intensity observed in the inter-chromosomal regions of the same nucleus. The average difference between axis-associated and inter-chromosomal signal for COH-3/4 was increased by 114% in *wapl-1* mutants compared with wild-type controls, while the increase observed for REC-8 and SMC-3 was 27% and 59% respectively (*Figure 5C*). The intermediate increase of axis-associated SMC-3 levels is to be expected since SMC-3 should form part of both REC-8 and COH-3/4 cohesin complexes. Separation of soluble and DNA-bound protein fractions from whole-worm extracts confirmed that the amount of COH-4 associated with DNA is increased in *wapl-1* mutants compared to wild-type controls (*Figure 5D–E* and *Figure 5—figure supplement 5*). Therefore, in the absence of WAPL-1 there is a large increase in the amount of cohesin associated with axial elements, and most of this increase corresponds to cohesin complexes containing the COH-3/4 kleisins.

## WAPL-1 antagonizes SCC mediated by COH-3/4 during late prophase

Following exit from pachytene, meiotic chromosomes undergo a condensation process that culminates with the formation of highly compacted and individualized chromatin structures seen in diakinesis oocytes, where 6 bivalents (pairs of homologous chromosomes linked together by chiasmata) are present in wild-type oocytes (*Figure 6A*). The presence of between 7 and 12 chromatin masses in diakinesis oocytes indicates defects in the formation of chiasmata, while defects in SCC typically result in oocytes containing more than 12 chromatin bodies (*Figure 6A*). Although gross loss of SCC in diakinesis oocytes requires the simultaneous depletion of REC-8, COH-3 and COH-4 (*Severson et al., 2009*; *Tzur et al., 2012*; *Severson and Meyer, 2014*), REC-8 appears to play a more prominent role in mediating SCC than COH-3/4. For example, removal of REC-8 from *spo-11* mutants, which fail to form meiotic DSBs, induces extensive separation of sister chromatids in oocytes, and removal of REC-8 from mutants lacking the SC results in oocytes with separated sisters

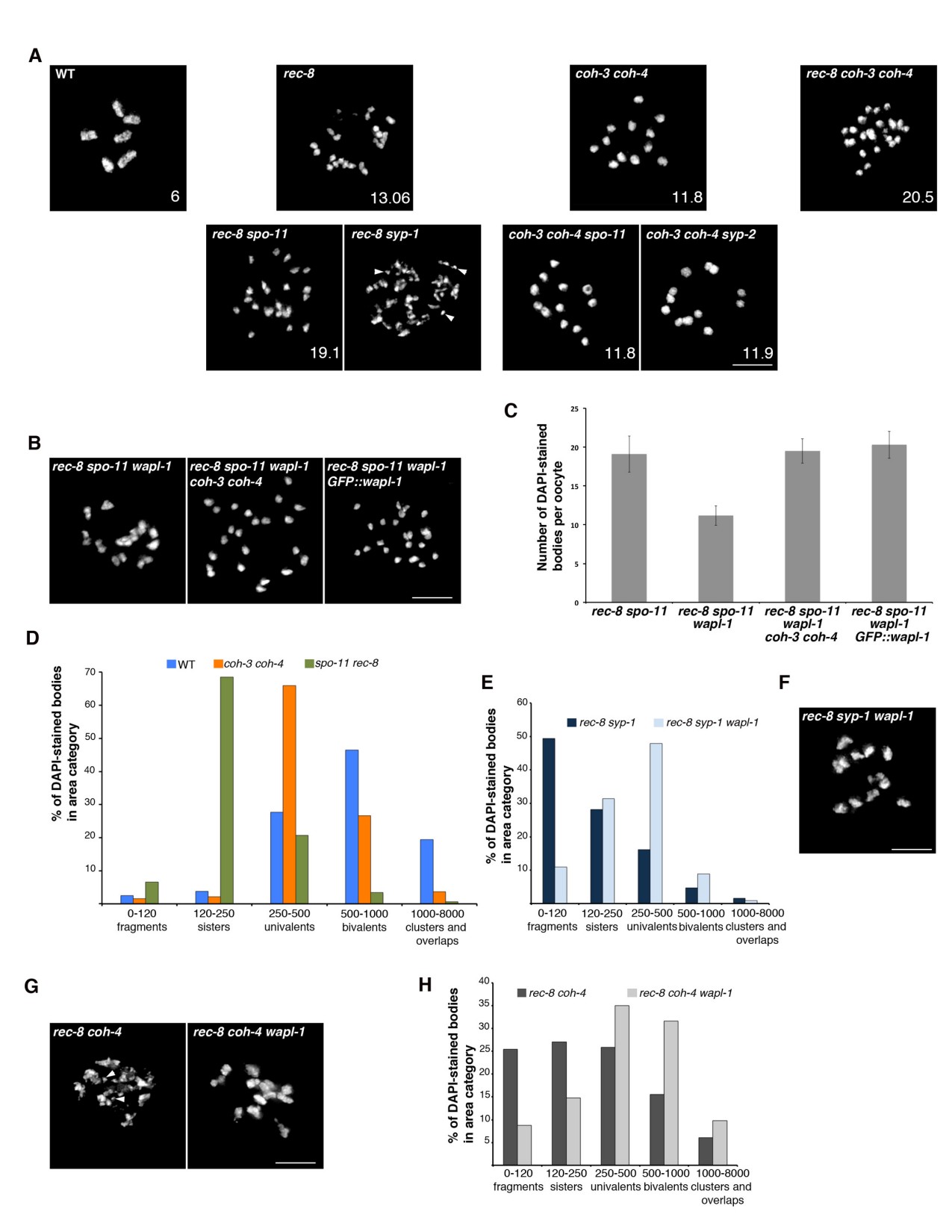

**Figure 6.** WAPL-1 antagonizes COH-3/4 cohesion in diakinesis oocytes. (**A**) Projections of diakinesis oocytes of the indicated genotypes stained with DAPI. 6 DAPI-stained bodies (WT) indicates presence of 6 bivalents, 12 DAPI-stained bodies indicates absence of chiasmata, and the presence of more

*Figure 6 continued on next page*

*Figure 6 continued*

than 12 DAPI-stained bodies indicates separation of sister chromatids, with 24 demonstrating separation of all sisters. The average number of DAPI-stained bodies observed in each genotype is indicated on the bottom right of each panel, number of nuclei analyzed: WT (20), *rec-8* (29), *coh-3 coh-4* (16), *rec-8 coh-3 coh-4* (17), *rec-8 spo-11* (94), *coh-3 coh-4 spo-11* (26), and *coh-3 coh-4 syp-2* (26). Arrowheads in *rec-8 syp-1* panel point to chromosome fragments. Note that removal of SPO-11 or SYP-1/2 causes loss of cohesion in *rec-8* mutants but not in *coh-3 coh-4* double mutants. (B) Projections of diakinesis oocytes of the indicated genotypes stained with DAPI, quantification shown in (C). Note the reduction of DAPI-stained bodies in *rec-8 spo-11 wapl-1* compared with three other genotypes (*p*<0.0001 in all cases, *t*-test). Error bars = standard deviation. Number of nuclei analyzed: *rec-8 spo-11* (94), *rec-8 spo-11 wapl-1* (85), *rec-8 spo-11 wapl-1 coh-3 coh-4* (29), *rec-8 spo-11 wapl-1GFP::wapl-1* (27). (D) Automated quantification (CellProfiler) of area sizes corresponding to chromatin bodies in projections of diakinesis oocytes of indicated genotypes stained with DAPI. Values on the X axis represent area in pixels and binning of the different categories was adjusted using oocytes of known phenotypes: WT (bivalents), *coh-3 coh-4* (univalents) *spo-11 rec-8* (detached sisters). Number of oocytes analyzed: WT (40), *coh-3 coh-4* (61) *spo-11 rec-8* (116). (E) Automated quantification of area sizes corresponding to chromatin bodies in projections of diakinesis oocytes, note that removing WAPL-1 from *rec-8 syp-1* double mutants causes a large decrease of chromosome fragments and an increase in univalents. Number of nuclei analyzed: *rec-8 syp-1* (52), *rec-8 syp-1 wapl-1* (36). (F) Example of DAPI-stained oocyte from *rec-8 syp-1 wapl-1* demonstrating absence of chromosome fragments, compare with *rec-8 syp-1* example shown in A. (G) Projections of diakinesis oocytes stained with DAPI, note that chromosome fragments are present in *rec-8 coh-4* (arrowheads) but not in *rec-8 coh-4 wapl-1* oocytes. (H) Automated quantification of area sizes corresponding to chromatin bodies in projections of diakinesis oocytes, note that removing WAPL-1 from *rec-8 coh-4* double mutants causes a large decrease of chromosome fragments. 26 diakinesis oocytes were analyzed for both *rec-8 coh-4* and *rec-8 coh-4 wapl-1*.

The following figure supplements are available for figure 6:

**Figure supplement 1.** Control demonstrating that unscheduled DSBs are not formed in *wapl-1 spo-11 rec-8* triple mutants.

**Figure supplement 2.** Examples of automated area analysis using CellProfiler.

and extensive chromosome fragmentation (*Figure 6A*) (*Colaiácovo et al., 2003*; *Severson et al., 2009*; *Severson and Meyer, 2014*). In contrast, removing SPO-11 (*Severson and Meyer, 2014*) or the SC from *coh-3 coh-4* double mutants does not compromise SCC in diakinesis oocytes (*Figure 6A*). These observations, together with the finding that WAPL-1 antagonizes the association of COH-3/4 with axial elements, led us to investigate if the antagonistic effect of WAPL-1 on COH-3/4 may be responsible for the compromised cohesion in mutants lacking REC-8. We first tested if removing WAPL-1 from *spo-11 rec-8* double mutants had an effect on SCC observed in oocytes. Strikingly, the presence of detached sister chromatids was dramatically reduced in the oocytes of *wapl-1 spo-11 rec-8* triple mutants, which displayed an average of 11.2 DAPI-stained bodies, while diakinesis oocytes of *spo-11 rec-8* double mutants displayed an average of 19.1 DAPI-stained bodies, demonstrating that WAPL-1 antagonizes cohesion in *spo-11 rec-8* oocytes (*Figure 6B–C*). Crucially, cohesion observed in *wapl-1 spo-11 rec-8* oocytes is mediated by COH-3/4, since sister chromatids were separated in oocytes of *wapl-1 spo-11 rec-8 coh-3 coh-4* quintuple mutants (*Figure 6B–C*). Given that DSBs are required for tethering sister chromatids in *rec-8* oocytes (*Severson and Meyer, 2014*), we tested if the cohesion rescue observed in *wapl-1 spo-11 rec-8* oocytes could be due to the presence of unscheduled DSBs in these triple mutants. However, RAD-51 foci were not observed in germ lines of *wapl-1 spo-11 rec-8* triple mutants (*Figure 6—figure supplement 1*), suggesting that removal of WAPL-1 increases COH-3/4-mediated cohesion independently of DSBs. Introducing the *GFP::wapl-1* transgene into *wapl-1 spo-11 rec-8* triple mutants resulted in oocytes with separated sister chromatids (*Figure 6B–C*), confirming that WAPL-1 antagonizes COH-3/4-mediated cohesion.

Next, we tested if WAPL-1 contributes to the chromosome fragmentation observed in oocytes of *rec-8 syp-1* double mutants. The extensive presence of small chromosome fragments with irregular shapes in *rec-8 syp-1* oocytes made manual quantification of the number of DAPI-stained bodies impractical. Thus, we used CellProfiler to determine the boundaries and area (in pixels) of each DAPI-stained body in projections of diakinesis oocytes. By imaging diakinesis oocytes from wild type, *coh-3 coh-4* double mutants and *rec-8 spo-11* double mutants we were able to calibrate the area range corresponding to bivalents, univalents and sister chromatids respectively. Despite the fact that some overlap of chromosomes occurs in projections of diakinesis oocytes, this automated method clearly identifies the predominant types of chromatin bodies known to be present in each of the three genotypes (*Figure 6D* and *Figure 6—figure supplement 2*). Based on this analysis,

chromosome fragments were defined by an area smaller than 120 pixels and 49% of chromatin bodies in *syp-1 rec-8* oocytes fell within this category (*Figure 6E*). In contrast, chromosome fragments only accounted for 10% of chromatin bodies in *wapl-1 rec-8 syp-1* oocytes (*Figure 6E–F*). This reduction in chromosome fragments was accompanied by a large increase in the number of chromatin bodies corresponding to univalents. Thus, WAPL-1 is largely responsible for the chromosome fragmentation and sister separation seen in *rec-8 syp-1* oocytes, likely by antagonizing COH-3/4 cohesin. We further tested this possibility by building a *coh-4 rec-8* double mutant, in which we hypothesized that the presence of a single meiotic kleisin (COH-3) should result in a worsening of the cohesion defects observed in *rec-8* single mutants. In agreement with this, 25% of chromatin bodies in *coh-4 rec-8* oocytes had an area corresponding to chromosome fragments, while this number was reduced to 8% in *wapl-1 coh-4 rec-8* oocytes (*Figure 6G–H*). These observations demonstrate that WAPL-1 antagonizes SCC mediated by COH-3/4 both in the presence and absence of SPO-11 DSBs.

## Inter-sister attachments in *rec-8* oocytes require recombination events promoted by the crossover pathway

Sister chromatids remain attached in the diakinesis oocytes of *rec-8* mutants, but the contact between them is often limited to a very small region, giving *rec-8* univalents a bilobed appearance that contrasts with the rounded appearance of univalents observed in recombination-deficient mutants such as *spo-11* (*Figure 6A*)(*Severson et al., 2009*; *Collette et al., 2011*). Why sisters are weakly attached in *rec-8* oocytes remains unknown, but these attachments require SPO-11 DSBs (*Severson et al., 2009*), leading to the proposal that DSBs induce phosphorylation of COH-3/4, which in turn make these complexes cohesive in a pathway similar to the damage-induced cohesion observed in mitotic yeast cells (*Heidinger-Pauli et al., 2008*; *Severson and Meyer, 2014*). However, we considered the possibility that the formation of inter-sister recombination events during the repair of SPO-11 DSBs leads to the formation of sister attachments in *rec-8* oocytes. To test this hypothesis we built worms lacking REC-8 and COSA-1, a protein required for the late stages of crossover formation but not for the induction of DSBs (*Yokoo et al., 2012*). Strikingly, oocytes from *rec-8 cosa-1* double mutants display a dramatic increase in the number of DAPI-stained bodies (average 23.89) compared with *rec-8* (average 13.06) and *cosa-1* (average 11.83) single mutants, suggesting extensive separation of sister chromatids (*Figure 7A–B*). FISH experiments confirmed extensive separation of sister chromatids in diakinesis oocytes of *rec-8 cosa-1* double mutants, while also demonstrating that SCC is not affected in pachytene nuclei of these same mutants (*Figure 7C–D*). Thus, loss of cohesion must occur during the chromosome remodeling process that starts at late pachytene. These results suggest that attachments between sister chromatids in diakinesis oocytes of *rec-8* mutants require the presence of a recombination event formed by the COSA-1-dependent crossover pathway. Moreover, similar to the situation in *spo-11 rec-8* oocytes, removing WAPL-1 from *rec-8 cosa-1* mutants restored cohesion in diakinesis oocytes (*Figure 7A–B*), confirming that WAPL-1 antagonizes cohesion mediated by COH-3/4 during late prophase.

## A WAPL-1-independent mechanism removes cohesin during late diakinesis

Imaging of germ lines from wild-type worms expressing SMC-1::GFP demonstrated that the soluble pool of SMC-1 that starts accumulating at late pachytene persists during diakinesis, and that the oocyte about to be ovulated (-1 oocyte) displayed a reduction in the intensity of chromosome-associated cohesin compared to the -2 oocyte of the same germ line (*Figure 8A*). Since Wapl is required for the prophase pathway that removes cohesin before metaphase in mitotic cells (*Haarhuis et al., 2014*) and a similar pathway operates during meiosis in plants (*De et al., 2014*), we tested whether WAPL-1 is required for cohesin removal in late diakinesis oocytes. Surprisingly, a clear reduction in chromosome-associated cohesin occurs in late diakinesis oocytes of *wapl-1* mutant germ lines (*Figure 8A*). In most cases (21 out of 26 analyzed germ lines) this reduction is evident between the -2 and -1 oocytes, while in a few germ lines the reduction was more evident between the -3 and -2 oocytes (*Figure 8C* and *Figure 8—figure supplement 1*). Similar to what we observed in diplotene, diakinesis oocytes of *wapl-1* mutants lack the accumulation of nuclear soluble SMC-1 observed in WT oocytes (*Figure 8A*). We also imaged diakinesis oocytes using super resolution structural illumination microscopy, which allowed us to observe in greater detail changes in chromosome-associated

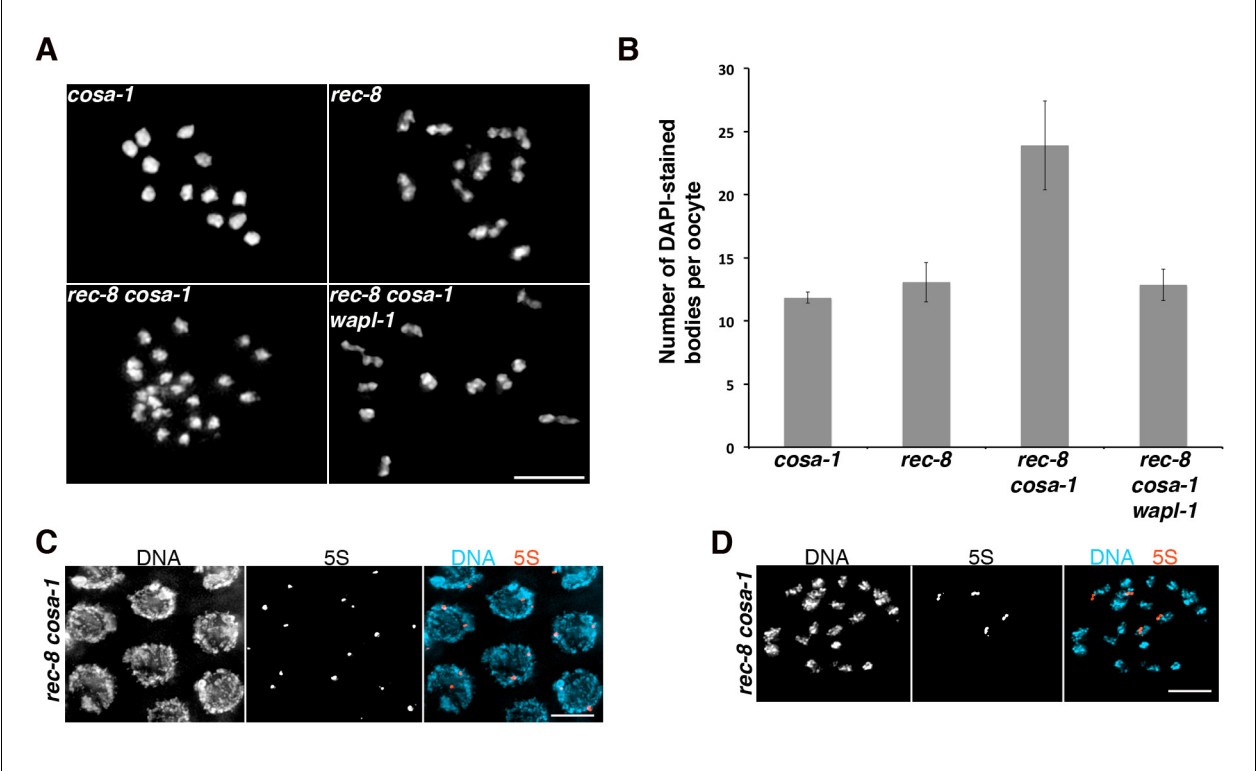

**Figure 7.** Tethering of sister chromatids in *rec-8* oocytes requires crossover precursors. (**A**) Projections of diakinesis oocytes stained with DAPI. Sisters are detached in *rec-8 cosa-1* oocytes, but not in *rec-8 cosa-1 wapl-1* oocytes. (**B**) Quantification of the number of DAPI-stained bodies in diakinesis oocytes of indicated genotype, note significant increase of DAPI-stained bodies *in rec-8 cosa-1* compared with three other genotypes ($p < 0.0001$ in all cases, *t*-test). Error bars= standard deviation. Number of nuclei analyzed: *cosa-1* (65), *rec-8* (29), *rec-8 cosa-1* (39), *rec-8 cosa-1 wapl-1* (42). (**C**) Projection of pachytene nuclei labeled by FISH with 5S rDNA probe and stained with DAPI. The presence of two signals per nucleus (one per homolog) indicates that sister chromatids are not separated. (**D**) Projection of a diakinesis oocyte labeled by FISH with 5S rDNA probe and stained with DAPI. The four signals indicate separation of sister chromatids. Scale bar in all panels = 5 µm.

cohesin. These experiments also demonstrated that a reduction in chromosome bound SMC-1::GFP occurs in late diakinesis oocytes of both wild-type and *wapl-1* mutant germ lines (*Figure 8B*). Thus, a WAPL-1-independent wave of cohesin removal occurs during late diakinesis.

## Discussion

Our study reveals that WAPL-1 antagonizes stable cohesin binding throughout meiotic prophase, affecting the morphogenesis of axial elements at the onset of meiotic prophase, the higher-order structure of chromosomes during pachytene, the process of chromosome remodeling during diplotene, and cohesion in diakinesis oocytes. Thus, WAPL-1 affects meiotic prophase events by promoting cohesin removal from the beginning of meiosis, not just as part of a meiotic prophase pathway that removes cohesin during the late stages of chromosome condensation in preparation for the first meiotic division. Importantly, we have uncovered that cohesin complexes containing the COH-3/4 kleisins, rather than REC-8, are the main targets of WAPL-1, showing that kleisin identity is key in determining the sensitivity of meiotic cohesin to WAPL-1 activity.

### WAPL-1 regulates meiotic chromosome structure

Similar to mitotic prophase chromosomes, meiotic chromosomes are organized as linear arrays of sister-chromatid loops that are attached at their base to a proteinaceous axial element containing cohesin (*Kleckner, 2006*; *Liang et al., 2015*). Under this organization, the shape of chromosomes is largely determined by the size of chromatin loops and by the compaction exerted by axial elements

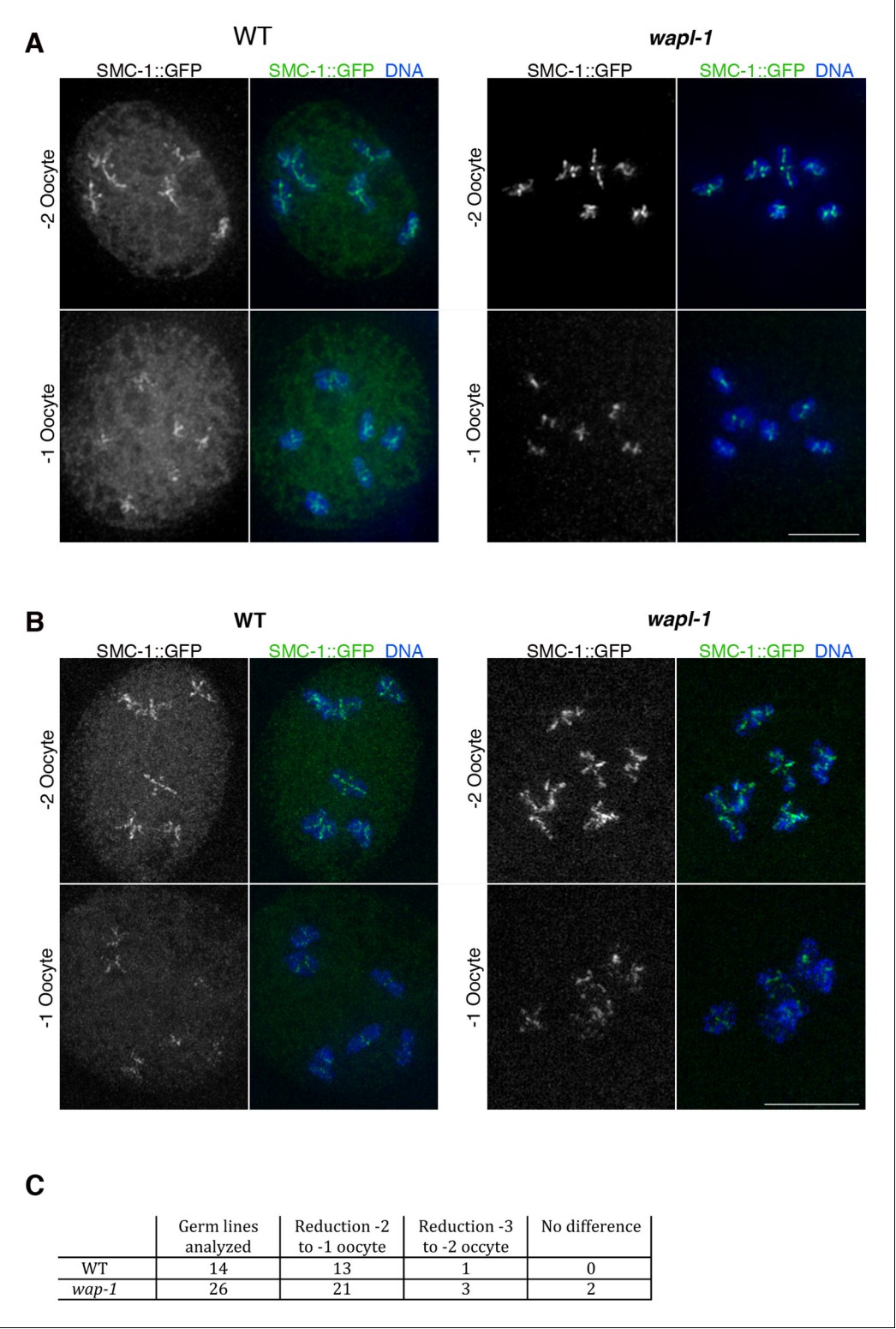

**Figure 8.** Cohesin is removed by a WAPL-1-independent mechanism in late diakinesis oocytes. Projections of diakinesis oocytes from worms expressing SMC-1::GFP (tagged by CRISPR) stained with anti-GFP antibodies and DAPI, and imaged with a Delta Vision system (**A**) or a structured illumination microscope (**B**). WT oocytes accumulate a large amount of nuclear soluble SMC-1::GFP that is lacking in *wapl-1* mutant oocytes, but a reduction in chromosome-associated SMC-1::GFP occurs in -1 oocytes of both WT and *wapl-1* mutants. -1 and -2 oocytes shown for each genotype were part of the same germ line and were acquired on the same image. Scale

*Figure 8 continued*

bars = 5 µm. (C) Table showing analysis of reduction in chromosome-associated SMC-1::GFP staining in late diakinesis oocytes of WT and *wapl-1* mutant worms.

The following figure supplement is available for figure 8:

**Figure supplement 1.** WAPL-1-independent reduction in SMC-1::GFP staining during late meiotic prophase.

on loop bases, two parameters that must be regulated by proteins localizing to axial elements. In fact, the meiosis-specific subunit Smc1β, which localizes to axial elements, regulates the organization of chromatin loops and the length of axial elements in mouse pachytene oocytes (*Novak et al., 2008*). This suggests that a key role of meiosis-specific cohesin complexes may be to modulate the higher-order organization of chromosomes. Our studies provide strong support for this hypothesis. First, the formation of SMC-1-labelled axial elements is observed in nuclei before the onset of meiotic prophase (leptotene) in *wapl-1* mutants, but not in wild-type germ lines. Interestingly, the intensity of GFP::WAPL-1 is at its highest in nuclei preceding leptotene, suggesting that WAPL-1 antagonizes cohesin binding from the onset of meiosis and that the regulation of WAPL-1 levels, and presumably its activity, may be coupled to meiotic progression. Second, in pachytene nuclei removal of WAPL-1 causes a dramatic increase in the levels of chromosome-associated COH-3/4 cohesin and shortening of axial elements, consistent with COH-3/4 complexes acting as regulators of axis compaction. Shortening of axial elements is also observed in yeast cells lacking *wpl1* (A. Shinohara personal communication), consistent with our findings. Interestingly, we observed that the shortening of axial elements caused by the absence of WAPL-1 is exacerbated in mutants lacking the SC, hinting that the SC is somehow capable of antagonizing cohesin's ability to induce compaction of axial elements.

In contrast to COH-3/4, REC-8 levels remain similar to wild-type controls throughout meiotic prophase of *wapl-1* mutant germ lines. The specific effect of WAPL-1 on COH-3/4 complexes is consistent with the finding that REC-8 and COH-3/4 display clear differences in their timing and mechanism of loading during early prophase (*Severson and Meyer, 2014*), and may help to explain some of the functional differences between REC-8 and COH-3/4 cohesin. For example, the fact that REC-8 complexes are refractory to the removal activity of WAPL-1 is consistent with these complexes playing a prominent role in providing cohesion, while COH-3/4 promote SC assembly in a cohesion-independent fashion (*Severson and Meyer, 2014*). Similarly, mouse Rad21L promotes homolog pairing without playing a direct role in cohesion, presumably by regulating higher-order structure of chromosomes during early prophase (*Ishiguro et al., 2014*). A clear precedent for cohesin affecting chromosome topology independently of cohesion is observed when Wapl is removed from mammalian cells before S-phase, which causes unscheduled condensation of unreplicated chromosomes (*Tedeschi et al., 2013*). We propose that by antagonizing the stable association of COH-3/4 cohesin to axial elements, WAPL-1 acts as a major regulator of meiotic chromosome structure.

## Why are COH-3/4, but not REC-8, complexes sensitive to WAPL-1?

The finding that WAPL-1 has a strong antagonistic effect on the chromosomal association of COH-3/4, but not REC-8, suggests that some property of REC-8 cohesin must make these complexes refractory to the removal activity of WAPL-1. In mitotic cells, Wapl is thought to mediate cohesin removal by triggering opening of an 'exit gate' present at the interface delimited by the interaction between the N-terminal region of Smc3 and the Scc1 kleisin (*Chan et al., 2012*; *Gligoris et al., 2014*; *Huis in 't Veld et al., 2014*). Opening of this interface by Wapl is antagonized by acetylation of two conserved lysines in Smc3, a process that is mediated by the acetyl transferase Eco1 (*Ben-Shahar et al., 2008*; *Unal et al., 2008*; *Zhang et al., 2008*). Under unchallenged growth conditions, Smc3 acetylation only occurs during DNA replication, since Eco1 is degraded after S-phase (*Lyons and Morgan, 2011*). Thus, cohesin that associates with chromosomes during G2 is not acetylated and shows a rapid turnover mediated by Wapl (*Kueng et al., 2006*). Interestingly, REC-8 is the only meiosis-specific kleisin subunit that appears to be loaded to chromosomes during S-phase in mice and worms, while Rad21L (mice) and COH-3/4 (worms) are loaded post S-phase (*Ishiguro et al., 2014*;

*Severson and Meyer, 2014*). Thus, a plausible explanation for our observations is that only REC-8 forms part of cohesin complexes in which SMC-3 is acetylated, while COH-3/4 associate with non-acetylated SMC-3 and therefore are sensitive to WAPL-1. However, mechanisms that protect cohesin from the removal activity of Wapl independently of Smc3 acetylation have been described in yeast and human mitotic cells. For example, DSBs induce the establishment of Wpl1-resistant cohesion during G2 by a mechanism involving Eco1-mediated acetylation of Scc1 (*Heidinger-Pauli et al., 2009*). Interestingly, depletion of Eco1 after meiotic S-phase induces chromosome segregation defects in *Drosophila* oocytes, suggesting that Eco1 is functional during meiotic prophase, although the targets of Eco1 in this case remain unknown (*Weng et al., 2014*). In addition to acetylation by Eco1, the presence of DNA damage also induces Scc1 sumoylation, which is required for establishment of cohesion and sister chromatid-mediated homologous repair during G2 (*McAleenan et al., 2012*; *Wu et al., 2012*). If posttranslational modifications on REC-8, or another subunit of REC-8 complexes, are required to antagonize the releasing activity of WAPL-1, untimely loss of these modifications could lead to premature release of REC-8 cohesin, compromising chromosome segregation during the meiotic divisions. Elucidating the molecular mechanisms that prevent removal of REC-8 complexes by WAPL-1 during meiotic prophase remains an important goal for future studies.

## Cohesion is modulated by WAPL-1 and recombination during late prophase chromosome remodeling

Our results indicate that cohesin removal by WAPL-1 is an important aspect of the chromosome remodeling process that starts during late pachytene and that involves SC disassembly and changes in chromosome condensation. In organisms as diverse as *Sordaria* and mouse (oocytes), visualization of cohesin during late prophase demonstrates that the emergence of compacted chromosomes attached by chiasmata is preceded by the large disappearance of axial elements as continuous linear structures, and by the accumulation of diffuse cohesin staining in the nucleus (*Prieto et al., 2004*; *Storlazzi et al., 2008*). We report that WAPL-1 promotes loosening of axial element structure and accumulation of soluble cohesin during late prophase in *C. elegans*, suggesting that a Wapl-dependent weakening of cohesin-mediated chromosome organization may be a general feature of late prophase chromosome remodeling. Importantly, this wave of WAPL-1-mediated cohesin removal has a direct impact on the ability of COH-3/4 complexes to provide cohesion following the completion of chromosome remodeling. In the absence of REC-8, attachment of sister chromatids in diakinesis oocytes requires COH-3/4 cohesin and SPO-11 (*Severson et al., 2009*), but since sister chromatids are attached in pachytene nuclei of *spo-11 rec-8* double mutants (*Severson and Meyer, 2014*), loss of cohesion must occur during the process of chromosome remodeling via an unknown mechanism. By showing that removal of WAPL-1 causes a rescue of cohesion in oocytes of *spo-11 rec-8* double mutants and that this rescue requires COH-3/4 cohesin, we clearly identify WAPL-1 as a factor that antagonizes COH-3/4-mediated cohesion during late prophase.

Our investigation of the mechanisms that mediate cohesion in diakinesis oocytes of *rec-8* mutants has uncovered an important role for recombination in the modulation of cohesion during late meiotic prophase. The requirement of DSBs for cohesion in diakinesis oocytes of *rec-8* mutants has been explained by the existence of a mechanism similar to the break-induced cohesion observed in mitotic yeast cells (*Heidinger-Pauli et al., 2008*; *Severson and Meyer, 2014*). Under this model, DSBs trigger establishment of COH-3/4 cohesion via a mechanism involving CHK-2-mediated phosphorylation of COH-3/4. However, we show that removal of COSA-1, a protein that localizes to chromosomes during late pachytene and that is not required for the formation and early processing of DSBs (*Yokoo et al., 2012*), causes loss of cohesion in diakinesis oocytes of *rec-8* mutants. Therefore, the presence of DSBs alone is not sufficient to induce COH-3/4 cohesion that persists in diakinesis oocytes. Instead, tethering of sister chromatids in *rec-8* mutant oocytes appears to require the presence of crossover-fated recombination events. This requirement would explain the lack of cohesion in diakinesis oocytes of *spo-11 rec-8*, which fail to initiate recombination, and in oocytes of *syp-1 rec-8* double mutants, since SYP-1 is required for crossover formation (*MacQueen et al., 2002*). Interestingly, SC proteins are thought to load between sister chromatids in mouse and worm mutants lacking REC-8 (*Xu et al., 2005*; *Rog and Dernburg, 2013*; *Severson and Meyer, 2014*), suggesting that SC-promoted recombination intermediates may form between sister chromatids in *rec-8* mutants. In agreement with this possibility, crossover designation appears to occur normally in *Sordaria rec8* mutants, which also display SC formation between sister chromatids (*Storlazzi et al.,*

*2008*). Since sister chromatids are not separated in pachytene nuclei of *spo-11 rec-8, syp-1 rec-8* (*Severson and Meyer, 2014*) or *cosa-1 rec-8* (this study), and removal of WAPL-1 from these three mutants restores cohesion in diakinesis oocytes, the antagonistic effect of WAPL-1 on COH-3/4 cohesion must occur after pachytene exit. Our studies identify chromosome remodeling as a key stage in the modulation of cohesion during meiotic prophase, and suggest that a complex interplay between late crossover precursors and WAPL-1 plays an important role in this process.

### Active cohesin removal during meiotic prophase and human aneuploidy

An essential aspect of meiotic chromosome segregation is ensuring that chiasmata remain intact until the onset of anaphase I. This issue is particularly relevant to human fertility, as oocytes establish chiasmata during fetal development, but then arrest at the dictyate stage, with the first meiotic division only taking place at ovulation, up to several decades later. In fact, cohesion deterioration in arrested oocytes has been proposed as a contributing factor to explain why the incidence of aneuploidy increases dramatically with maternal age (*Nagaoka et al., 2012*; *Herbert et al., 2015*). Data from different mouse models offer support for this possibility. First, partial loss of cohesin before metaphase I in oocytes of mice lacking Smc1β (a meiosis-specific Smc1 subunit) causes the dissolution of chiasmata (*Hodges et al., 2005*). Second, Rec8-mediated cohesion is not regenerated during late stages of meiotic prophase (*Tachibana-Konwalski et al., 2010*) and Smc1β expressed before birth is sufficient to ensure chiasma maintenance throughout adult age (*Revenkova et al., 2010*). Third, age-related deterioration of bivalents is observed in oocytes from *Smc1β* mutant mice (*Revenkova et al., 2010*), as well as in oocytes from old (long lived) wild-type mice that also display defects in chromosome segregation and depletion of cohesin from chromosomes before anaphase onset (*Lister et al., 2010*). These observations are consistent with the view that cohesin deterioration could be an important contributor to human aneuploidy. However, the mechanisms responsible for cohesin depletion in aged oocytes are not known, with possible explanations including decay of cohesin molecules over time and leaky separase activity (*Jessberger, 2012*). We have found that two different mechanisms actively remove cohesin during late prophase as part of normal meiotic progression: First, WAPL-1 antagonizes cohesin during chromosome remodeling and second, a WAPL-1-independent wave of cohesin removal occurs during oocyte maturation. These findings could be relevant to human aneuploidy, as excessive cohesin removal by the WAPL-1-dependent or independent mechanisms identified here could compromise maintenance of chiasmata before the onset of anaphase I, inducing errors in chromosome segregation during the meiotic divisions.

## Materials and methods

### *C. elegans* genetics

All strains were maintained at 20°C and the N2 strain was used as the wild-type control. The following mutant alleles were used: *wapl-1(tm1814)*, *syp-1(me17)*, *rec-8(ok978)*, *coh-3(gk112)*, *coh-4 (tm1857)*, *spo-11(ok79)*, *cosa-1(tm3298)*, *syp-2 (ok307)*. The following transgenes were used: *meIs8 [unc-119(+) pie-1^{promoter}::GFP::cosa-1]* (*Yokoo et al., 2012*), *[unc-119(+) pie-1^{promoter}::zhp-3::GFP]* (*Jantsch et al., 2004*), *itIs37[unc-119(+) pie-1^{promoter}::mcherry::his-58]*.

Transgenic strains carrying single copy insertions of the *GFP::wapl-1* and *rec-8::GFP* transgenes in the *ttTi5605* locus on chromosome II were created following the protocol described in (*Frøkjær-Jensen et al., 2008*). The *GFP::wapl-1* transgene carried 494 bp upstream of the starting codon, a GFP cDNA containing 3 artificial introns and the entire sequence of the *wapl-1* locus plus 1505 bp of downstream sequence. The *rec-8::GFP::* transgene carried 536 bp upstream of the starting codon, the entire sequence of the *rec-8* locus, a GFP cDNA containing 3 artificial introns, and 665 bp of downstream sequence. Tagging of the endogenous *smc-1* locus by CRISPR was performed with an sgRNA targeting the 'GTTGCAATCGATGGTGTTGG' sequence at the 3' end of *smc-1* and a repair template containing GFP cDNA with 3 artificial introns flanked by 1400 bp of upstream and downstream sequence from the site of the DSB. Expression of sgRNA and Cas9 was performed using the protocols described in (*Friedland et al., 2013*).

### Immunostaining and FISH

Germ lines from young adult hermaphrodites were dissected, fixed and processed for immunostaining and FISH as described in (*Martinez-Perez and Villeneuve, 2005*). All images were acquired using a Delta Vision Deconvolution system equipped with an Olympus 1X70 microscope. Primary antibodies: rabbit anti-COH-3/4 were generated against residues 286-338 of the COH-3 protein; rabbit anti-HTP-1/2 were generated against residues 37-93 of HTP-1; chicken anti-SYP-1 were generated against a peptide including the first 23 residues of SYP-1; rabbit anti-HTP-3 (*Goodyer et al., 2008*); rabbit anti-RAD-51, anti-HIM-8, anti-REC-8 and anti-COH-3 antibodies were all purchased from Novus Biologicals; rabbit anti-SMC-3 (Millipore AB3914); rabbit anti-GFP conjugated to Alexa488 (Invitrogen).

### Quantitative analysis of cohesin staining

Mean whole nuclear fluorescence was quantified with an ImageJ macro written by D. Dormann and K. Hng. Quantification was performed on unprocessed raw images acquired as three-dimensional stacks using identical exposure settings in a Delta Vision system. Briefly, an oval defining the area of a nucleus was drawn and the mean fluorescence intensity within that area was quantified on each Z section and automatically averaged across the entire stack.

Fluorescence intensity line profiles were calculated on maximum intensity projections from unprocessed raw images acquired using identical exposure settings in a Delta Vision system using the SoftWoRx Line Profile tool. Lines were drawn to intersect with at least three axial elements and peak to trough fluorescence intensity ranges ($\Delta F$) were calculated for all peaks within a nucleus using DAPI signal as the reference for the position of chromosomes.

### Quantification of chromatin body area sizes in diakinesis oocytes

Images of DAPI-stained -1 and -2 diakinesis oocytes were acquired using a Delta Vision system equipped with an Olympus 1X70 microscope. Deconvolved image stacks were either used directly for visual counting, or were converted into maximum intensity projections, cropped to the size of an individual nucleus using ImageJ and analysed in CellProfiler to define the boundaries of each DAPI-stained body and to calculate its area in pixels. Following analysis of oocytes from control genotypes (*Figure 5D*), the following areas were used to classify DAPI-stained bodies: bivalents 500–1000 pixels, univalents 250–500 pixels, sister chromatids 120–250 pixels. DAPI-stained bodies with an area smaller than 120 pixels were counted as chromosome fragments.

### Super-resolution structured illumination microscopy

Immunostaining was performed as described above, but slides were mounted using ProLong Gold mounting media instead of vectashield. Images were acquired using a Zeiss Elyra microscope.

### In vivo imaging of embryos

24 hrs post L4 hermaphrodites were dissected to release embryos in a drop of 60% v/v Leibowitz-15 media, 20% fetal bovine serum, 25 mM HEPES pH 7.4, 5 mg/ml inulin. Embryos were then mounted on 2% agarose pads and imaged with a Delta Vision system equipped with an Olympus 1X70 microscope. Images of the meiotic divisions were acquired as series of 1 µm-spaced Z stacks (9–12 section) with a regular time lapse of 5 s intervals. Videos of these time series were created using SoftWoRx.

### Preparation of whole-worm protein extracts and Western blotting

100 young hermaphrodites were collected in 1X TE (10 mM Tris-HCl pH 8, 1 mM EDTA) supplied with complete protease inhibitor (Roche) and freeze-thawed three times in liquid nitrogen. Worms were then resuspended in 40 µl of 1X Laemli buffer and boiled for 10 min. Extracts were run on a 10% acrylamide gel and transferred on nitrocellulose for 1 hr at 4°C, blocked for 1 hour in 5% milk TBST (1x TBS 0.1% Tween) and then incubated with anti-WAPL-1 (1:3000) and goat anti-actin (Santa Cruz, 1:3000) antibodies. Anti-WAPL-1 antibodies were generated against residues 2-101 of the WAPL-1 protein.

## Protein fractionation from whole-worm extracts

In order to separate soluble and DNA-bound protein fractions from whole-worm extracts we followed the methods described in (*Silva et al., 2014*). Briefly, 150 young hermaphrodite worms were picked into a 1.5 ml tube containing 40 µl of extraction buffer (10 mM Tris-HCl pH 7.5, 1 mM EDTA, 2X Complete Protease Inhibitor and 2X Phospho-STOP), snap-frozen in liquid nitrogen, thawed, and grinded with a plastic pestle. Triton-X was added to a final concentration of 0.25% and tubes were placed on a thermomixer at 4°C in mild shaking (750 rpm) for 20 min. Tubes were then centrifuged for 10 min at 14,800 rpm at 4°C and the supernatant from this step, which represents the soluble fraction, was collected into a new tube and centrifuged once more under the same conditions to remove any debris. The remaining pellet, containing the non-soluble fraction, was resuspended with 40 µl of extraction buffer and washed twice before being resuspended in a final volume of 40 µl of extraction buffer. Laemmli buffer was added to a 1X final concentration and equal volumes of the protein extracts were run on a 7.5% acrylamide gel and transferred on nitrocellulose for 1 hr at 4°C, blocked for 1 hr in 5% milk TBST (1x TBS 0.1% Tween) and then incubated with the following primary antibodies over night at 4°C: rabbit anti-COH-4 (1:1000) (antibodies raised against residues 289–341 of the COH-4 protein), rabbit anti-HAL-2 (1:10,000) (*Zhang et al., 2012*), and rabbit anti-Histone H3 (1:10,000, AbCam). The following HRP-conjugated secondary antibodies were used in 5% milk in TBST for one hour at room temperature: donkey anti-goat (1:8000, Sigma) and goat anti-rabbit (1:5000, Millipore). Quantification of band intensities was performed using ImageJ software.

## Acknowledgements

We thank D. Dormann, K. Hng and C. Whilding from the MRC CSC microscopy facility for help with analysis of cohesin staining and with the structured illumination super resolution microscope. We are grateful to Akira Shinohara for sharing unpublished results. Research in the E.M-P laboratory is funded by a MRC core-funded grant.

## Additional information

### Funding

| Funder | Grant reference number | Author |
| --- | --- | --- |
| Medical Research Council | MC-A652-5PY60 | Enrique Martinez-Perez |

The funders had no role in study design, data collection and interpretation, or the decision to submit the work for publication.

### Author contributions

OC, Conception and design, Acquisition of data, Analysis and interpretation of data, Drafting or revising the article; CB, NF, NS, Acquisition of data, Analysis and interpretation of data, Drafting or revising the article; ST, MCP, ALJT, Acquisition of data, Analysis and interpretation of data; EMP, Conception and design, Analysis and interpretation of data, Drafting or revising the article

### Author ORCIDs

Oliver Crawley, http://orcid.org/0000-0002-5054-0051

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
