## [Decision Letter]

Thank you for submitting your work entitled "WAPL-1 regulates meiotic chromosome structure and cohesion by antagonizing specific cohesin complexes" for peer review at *eLife*. Your submission has been favorably evaluated by Jim Kadonaga (Senior Editor), a Reviewing editor, and two reviewers.

The reviewers have discussed the reviews with one another and the Reviewing editor has drafted this decision to help you prepare a revised submission.

Both reviewers felt that the manuscript presented some important new information, albeit not totally unexpected. However, several further experiments are required to solidify the paper:

Essential requirements:

1) Quantification/numbers analysed for Figure 1, Figure 3, Figure 5 must be provided.

2) A proper analysis of non-disjunction events in single WAPL-1 mutants.

3) The reviewers believe that the work is descriptive and would be helped significantly by a molecular analysis showing increased cohesion association with chromatin in WAPL-1 mutants. If this is not feasible, a deeper phenotypic analysis that gives more insight would bring this work to a more meaningful conclusion.

Other reviewer comments that would improve the paper:

Figure 3, figure legend: It is not stated in the legend what the arrowheads are pointing to in Figure 3. "While SMC-1::GFP tracks preleptotene nuclei of WAPL-1 mutants, in which DAPI staining also shows increased condensation". To me, the chromosomes in the WAPL-1 mutant look less condensed rather than more. Furthermore, there is no accompanying description/quantitation in the Results section of this important observation. A key point of the manuscript is the effect of WAPL-1 on meiotic chromosome structure, so this needs strengthening.

Figure legend 3F: "Note the axial elements become elongated and, twisted with a more diffuse appearance in wild-type nuclei compared to wapl-1 mutant nuclei." The twisting is difficult to see so should be pointed out as in an enlarged inset.

Figure 4: The figure (A) qualitatively shows the increased axial staining of COH-3/4 and the fluorescent intensity calculations (B) appear to be total rather than DNA specific. It would be more informative to quantitate the amount bound to DNA given the authors are trying to show an increased level of chromosome bound COH-3/4 in WAPL-1 mutants. This could be shown as the relative intensity/DAPI or the DAPI signal could be used as a mark to only calculate what is chromosome bound. […]

The assay is explained in the legend for Figure 5 ("6 DAPI-stained bodies (WT) indicates the presence of 6 bivalents etc"), however some accompanying explanation in the Results section would help the reader as I suspect most would be more familiar with chromosome loss or uneven segregation assays in meiosis.

The effect of the *wapl-1* mutation on chromosome compaction is not very convincing. The argument for this is based on cytological images and measurements of axes lengths. The measurement of axis lengths is confounded by chromosome overlap. From the images presented (Figure 3) it is not obvious that axes are shorter. In the measurements (Figure 3), the effect is subtle. Is it statistically significant? A better measure of chromosome compaction would be to look at distinct loci on a single chromosome by FISH.

In Figure 3, are the 5S rDNA foci further apart in the *wapl-1* cells?

---

## [Author Response]

*Essential requirements:*

*1) Quantification/numbers analysed for Figure 1, Figure 3, Figure 5 must be provided.*

We have added the following quantifications:

Figure 1: We have quantified the mean intensity of the GFP::WAPL-1 in premeiotic nuclei, transition zone nuclei (leptotene/zygotene), and in late pachytene nuclei. Differences in GFP::WAPL-1 intensity between premeiotic and leptotene nuclei, and between leptotene and late pachytene nuclei are significant. Graph showing this quantification is included in Figure 1—figure supplement 1.

Figure 3 (now Figure 4): Short axial elements corresponding to the X chromosomes (as indicated by HIM-8 staining) that can be traced along their entire length in projections of late pachytene nuclei are found in 3 out of 3 *syp-1 wapl-1* double mutant germ lines. In contrast, X chromosome axial elements cannot be traced on projections of late pachytene nuclei from syp-1 mutants due to extensive overlap of axial elements. We now indicate this on the figure legend of Figure 4, and we have also added a figure with a *syp-1* and a *wapl-1 syp-1* example (Figure 4—figure supplement 3).

Figure 3 (now Figure 4): We have performed new experiments to investigate the distribution of SMC-1::GFP at the onset of meiotic prophase in WT and *wapl-1* mutants. Specifically, we have used PLK-2 antibodies to clearly delimit the onset of meiotic prophase. This analysis has confirmed that tracks of SMC-1::GFP in nuclei preceding the onset of meiotic prophase are only observed in wapl-1 mutants. We now indicate in the main text the number of germ lines analyzed (8 in both WT and *wapl-1* mutants) and the average number of pre-leptotene nuclei displaying SMC-1::GFP tracks per germ line. The new data including PLK-2 staining is shown in Figure 4. We have also added a new figure showing SMC-1::GFP staining in nuclei spanning the stages between preleptotene and early pachytene in WT and wapl-1 mutant germ lines, as images containing this section of the germ line (much larger than that shown in Figure 4) help to visualize the differences in axial element organization between WT and wapl-1 mutants (Figure 4—figure supplement 4).

Figure 3 (now Figure 4): We have quantified the average intensity of nuclear soluble SMC-1::GFP in diplotene nuclei, confirming that signal intensity is significantly higher in WT nuclei than in wapl-1 mutant nuclei. This quantification is shown in Figure 4—figure supplement 5.

Figure 5 (now Figure 6): We have added the average number of DAPI-stained bodies observed in each genotype to the panels showing diakinesis oocytes in Figure 6. The number of oocytes analyzed per genotype is indicated in the figure legend.

Figure 7 (now Figure 8). We have added a table to Figure 8 (panel C) in which we show the analysis of SMC-1::GFP reduction during late diakinesis in 14 WT germ lines and 26 wapl-1 mutant germ lines. In addition, we have added a further example of the distribution of SMC-1::GFP during late diakinesis (Figure 8—figure supplement 1).

*2) A proper analysis of non-disjunction events in single WAPL-1 mutants.*

We have taken two complementary approaches to investigate if defects in chromosome segregation during the meiotic divisions occur in *wapl-1* mutant embryos. First, we have significantly extended our in vivo analysis of meiotic chromosome segregation by filming 11 *wapl-1* mutant and 14 wild type embryos. This new data demonstrates that the second polar body fails to be extruded in most (7/11) *wapl-1* mutant embryos, confirming our previous results on fixed embryos. Crucially, the new in vivo data reveals that the second polar body undergoes cycles of chromatin condensation and decondensation, mimicking the cell cycle of the oocyte pronucleus first and then mitotic nuclei, and that eventually chromosomes in the second polar body can fuse with the content of a mitotic nucleus. This demonstrates that the defects in polar body extrusion observed in *wapl-1* mutants induce the formation of embryos carrying abnormalities in chromosome number, likely contributing to the embryonic lethality that we report. This data is presented on a new Figure 3, which includes a table summarizing the findings from all filmed embryos, as well as in 3 supplemental Movies. Second, we have used FISH to directly assess the segregation of chromosome V during the meiotic divisions. This analysis in 17 WT and 17 *wapl-1* mutant embryos shows that all analyzed oocyte pronuclei contained a single copy of chromosome V, demonstrating normal segregation of chromosome V during the meiotic divisions. Although this FISH analysis cannot rule out segregation defects of the other five chromosomes present in *C. elegans*, our in vivo data clearly demonstrates that wapl-1 mutant embryos are defective in the extrusion of the second polar body and that this defect contributes to produce chromosomally abnormal embryos.

*3) The reviewers believe that the work is descriptive and would be helped significantly by a molecular analysis showing increased cohesion association with chromatin in WAPL-1 mutants. If this is not feasible, a deeper phenotypic analysis that gives more insight would bring this work to a more meaningful conclusion.*

We have performed protein fractionation experiments to investigate the subcellular localization of COH-4 in whole-worm extracts prepared from WT controls and *wapl-1* mutants. These new experiments demonstrate that in the absence of WAPL-1 most COH-4 is associated with DNA, while in its presence the amount of COH-4 in the soluble fraction is higher than in the DNA-bound protein fraction. These new experiments confirm that WAPL-1 restricts COH-4 binding to chromosomes, fully supporting our previous observations based on the analysis of cohesin distribution by immunostaining. We have also attempted to analyze the distribution of REC-8 by protein fractionation, but unfortunately available anti-REC-8 antibodies do not work well on western blots. The new data is presented in panels D and E of Figure 5, with additional controls shown in Figure 5—figure supplement 5.

*Other reviewer comments that would improve the paper: Figure legend 3F "Note the axial elements become elongated and, twisted with a more diffuse appearance in wild-type nuclei compared to* wapl-1

*mutant nuclei." The twisting is difficult to see so should be pointed out as in an enlarged inset.*

We have included insets in the SMC-1::GFP panels of Figure 4 to magnify specific chromosomal regions from diplotene nuclei of WT and *wapl-1* mutants, facilitating the comparison of axial element organization in the two genotypes.

Figure 4: The figure (A) qualitatively shows the increased axial staining of COH-3/4 and the fluorescent intensity calculations (B) appear to be total rather than DNA specific. It would be more informative to quantitate the amount bound to DNA given the authors are trying to show an increased level of chromosome bound COH-3/4 in WAPL-1 mutants. This could be shown as the relative intensity/DAPI or the DAPI signal could be used as a mark to only calculate what is chromosome bound. […]

We agree with this point. The quantification in Figure 5 (previously 4C) shows the amount of chromosome-associated cohesin. As shown on the top left panel of the figure, we used DAPI intensity to identify the position of chromosomes within the nucleus and then measured the value of cohesin subunits (COH-3/4, REC-8, or SMC-3) on chromosomal regions, as well as in regions lacking DAPI signal (inter chromosomal domains). Using this method we calculated the increment of cohesin staining specifically associated with chromosomes.

*The assay is explained in the legend for Figure 5 ("6 DAPI-stained bodies (WT) indicates the presence of 6 bivalents etc"), however some accompanying explanation in the Results section would help the reader as I suspect most would be more familiar with chromosome loss or uneven segregation assays in meiosis.*

We have added an explanation on the main text to clarify how the number of DAPI-stained bodies in diakinesis oocytes can be used to determine defects in cohesion (subsection “WAPL-1 antagonizes SCC mediated by COH-3/4 during late prophase”).

*The effect of the* wapl-1 *mutation on chromosome compaction is not very convincing. The argument for this is based on cytological images and measurements of axes lengths. The measurement of axis lengths is confounded by chromosome overlap. From the images presented (Figure 3) it is not obvious that axes are shorter. In the measurements (Figure 3), the effect is subtle. Is it statistically significant? A better measure of chromosome compaction would be to look at distinct loci on a single chromosome by FISH.*

Although chromosome overlaps occur in projections of pachytene nuclei stained with HTP-3 antibodies, the goal of this experiment was not be able to trace the six (paired) axial elements corresponding to the six pairs of homologs along their entire lengths, but rather to calculate the total length of HTP-3 tracks per nucleus, which can be easily obtained despite some overlaps. The differences in total HTP-3 length per nucleus between WT and *wapl-1* mutants are statistically significant (indicated in figure legend). Of note, chromosome overlap is much higher in projections of WT pachytene nuclei than in projections from *wapl-1* mutant pachytene nuclei, where it is often possible to trace 6 individual linear tracks (each corresponding o a pair of homologous chromosomes) (see also new Figure 4—figure supplement 4).

*In Figure 3, are the 5S rDNA foci further apart in the wapl-1 cells?*

Images shown are projections of 3D stacks and therefore calculating distances between signals corresponding to homologous chromosomes would require taking into account distances on Z the axis. While the data clearly shows that homologous chromosomes are separated in most pachytene nuclei of both *syp-1* and *syp-1 wapl-1* double mutants (see quantification in Figure 4—figure supplement 2), we have not attempted to quantify absolute distances between 5S rDNA signals on pachytene nuclei.